

# Deep learning methods for inverse problems

Shima Kamyab[1], Zohreh Azimifar[1,2], Rasool Sabzi[1] and Paul Fieguth[2]

[1] Department of Computer Science and Engineering, Shiraz University, Shiraz, Fars, Iran
[2] Department of Systems Design Engineering, University of Waterloo, Waterloo, Ontario, Canada

## ABSTRACT

In this paper we investigate a variety of deep learning strategies for solving inverse problems. We classify existing deep learning solutions for inverse problems into three categories of Direct Mapping, Data Consistency Optimizer, and Deep Regularizer. We choose a sample of each inverse problem type, so as to compare the robustness of the three categories, and report a statistical analysis of their differences. We perform extensive experiments on the classic problem of linear regression and three well-known inverse problems in computer vision, namely image denoising, 3D human face inverse rendering, and object tracking, in presence of noise and outliers, are selected as representative prototypes for each class of inverse problems. The overall results and the statistical analyses show that the solution categories have a robustness behaviour dependent on the type of inverse problem domain, and specifically dependent on whether or not the problem includes measurement outliers. Based on our experimental results, we conclude by proposing the most robust solution category for each inverse problem class.

## INTRODUCTION

An inverse problem (*Bertero & Boccacci, 1998*; *Fieguth, 2010*; *Stuart, 2010*) seeks to formulate a solution to estimating the unknown state underlying a measured system. Specifically, a forward function $F(\cdot)$ describes the relationship of a measured output $\underline{m}$,

$$\underline{m} = F(\underline{z}) + \underline{v} \tag{1}$$

as a function of the system state $\underline{z}$, subject to a measurement noise $\underline{v}$. The objective of the inverse problem is to estimate $\underline{z}$ as a function of given measurement $\underline{m}$, assuming a detailed knowledge of the system, $F(\cdot)$, where if $F(\cdot)$ is not known or is partially known the problem becomes *blind* or *semi-blind* (*Lucas et al., 2018*).

Different perspectives lead to different types of inverse problems. From the perspective of data type, two classes of inverse problems are *restoration* and *reconstruction* (*Arridge et al., 2019*), where restoration problems have the same domain for measurement and state (*e.g.*, signal or image denoising), while reconstruction has different domains (*e.g.*, 3D shape inference). Next, from the perspective of modeling, inverse problems are classified into *static* and *dynamic* problems, where the static case seeks a single estimate $\hat{\underline{z}}$, consistent with some prior model on $\underline{z}$ and the forward model $F(\underline{z})$, whereas the dynamic

Corresponding author
Zohreh Azimifar,
azimifar@cse.shirazu.ac.ir

case seeks estimates $\hat{\underline{z}}(t)$ over time, consistent with an initial prior and a dynamic model. We also consider a new class of inverse problems with some knowledge provided in the form of PDEs. In this paper we will examine each of these inverse problems.

Existing analytical methods for solving inverse problems take advantage of domain knowledge to regularize and constrain the problem to obtain numerically-stable solutions. These methods are classified into four categories in *Arridge et al. (2019)*:

- **Analytic inversion** (*Natterer, 2001*; *Schuster, 2007*) having the objective of finding a closed form, possibly approximate, of $F^{-1}$. This category of solutions will be highly problem dependent.
- **Iterative methods** (*Calvetti, Lewis & Reichel, 2002*; *Byrne, 2008*), which optimize the data consistency term

$$\min_{\underline{z}} ||\underline{m} - F(\underline{z})||. \tag{2}$$

Because of the ill-posed nature of most inverse problems, the iteration tends to have a semi-convergent behaviour, with the reconstruction error decreasing until some point and then diverging, necessitating appropriate stopping criteria.

- **Discretization as regularization** (*Hämarik et al., 2016*; *Kaltenbacher, Kirchner & Vexler, 2011*), including projection methods searching for an approximate solution of an inverse problems in a predefined subspace. Choosing an appropriate subspace has high impact on finding stable solutions.
- **Variational methods**, with the idea of minimizing data consistency penalized using some regularizer $R$ parameterized by $\theta$:

$$\min_{\underline{z}} ||\underline{m} - F(\underline{z})|| + R(\underline{z}, \theta) \tag{3}$$

This is a generic adaptable framework where $F(\cdot)$, $R(\cdot, \cdot)$ are chosen to fit a specific problem, of which well-known classical examples include (Tikhonov *Groetsch, 1984*) and total variation (*Makovetskii, Voronin & Kober, 2015*) regularization.

These approaches have weaknesses in requiring explicitly identified prior knowledge, selected regularizers, some shortcomings in handling noise, computational complexity in inference due to the optimization-based mechanisms, and most significantly limited applicability, in the sense that each inverse problem needs to be solved one-off.

As a result, we are highly motivated to consider the roles of Deep Neural Networks (DNNs) (*Larochelle et al., 2009*), which have the advantages of being generic data driven methods, are adaptable to a wide variety of different problems, and can learn prior models implicitly through examples. DNNs are currently in widespread use to solve a vast range of problems in machine learning (*Balas et al., 2019*), artificial intelligence (*Samek, Wiegand & Müller, 2017*), and computer vision *Kim et al. (2018)*. The strong advantages of using such structures include their near-universal applicability, their real-time inference

(*Canziani, Paszke & Culurciello, 2016*; *Khan et al., 2019*), and their superiority in handling sensor and/or measurement noise (*Han et al., 2018*).

A variety of studies (*Aggarwal, Mani & Jacob, 2018*; *Lucas et al., 2018*) have shown that planned, systematic DNNs will tend to have fewer parameters and better generalization power compared to generic architectures, which motivates us to consider systematic strategies in addressing complex inverse problems.

In principle, *every* deep learning framework could be interpreted as solving some sort of inverse problem, in the sense that the network is trained to take measurements and to infer, from given ground truth, the desired unknown state. For example, for the common DNN application to image classification, the input is a (measured) image, and the network output is a (unknown state) label, describing the object or scene appearing in the image. The network parameters then implicitly learn the inverse of the forward model, which had been the generation of an image from a label.

Using DNNs for solving inverse problems aims to approximate the inverse of the forward model (*Fieguth, 2010*). In some cases, the forward model may be explicitly defined (*Anirudh et al., 2018*; *Chang et al., 2017*; *Aggarwal, Mani & Jacob, 2018*), whereas in other cases it may be implicitly defined in the form of the training data (*Adler & Öktem, 2017*; *Antholzer, Haltmeier & Schwab, 2019*; *Jin et al., 2017*; *Kelly, Matthews & Anastasio, 2017*; *Anirudh et al., 2018*; *Zhang & Ghanem, 2018*; *Fan et al., 2017*). In this paper our focus is on solving *non-blind* inverse problems, with the forward model known. Analytical approaches to inverse problems, whether deterministic or stochastic, take advantage of the explicit forward model and prior knowledge in formulating the solution; in contrast, DNNs cannot take advantage of such information, and must instead learn implicitly from large datasets of training data in a black-box approach.

Inspired by the above techniques, there are indeed a number of proposed deep frameworks in the literature with the aim of bringing regularization techniques or prior knowledge into the DNN learning process for solving inverse problems (*Aggarwal, Mani & Jacob, 2018*; *Chang et al., 2017*; *Dosovitskiy et al., 2015*; *Wang et al., 2015*; *Xu et al., 2014*; *Schuler et al., 2015*; *Raissi, Perdikaris & Karniadakis, 2019*; *Bu & Karpatne, 2021*). In this paper, we classify deep solutions for inverse problems into four categories based on their objective criteria, and compare them in solving different types of inverse problems.

The focus of this paper is comparing the robustness of different deep learning structures based on their optimization criterion associated with the training scheme; that is, the main objective of this research is to provide insight into the choice of appropriate framework, particularly with regards to performance robustness. It is worth noticing here that our goal is not to outperform the state-of-the-art performance in different problems, nor to propose new deep-learning approaches, rather to examine different frameworks with fair parameter settings. Using these frameworks, we select a prototype inverse problem from each category and evaluate the performance and the robustness of the designed frameworks. We believe the results obtained in this way give insight into the strength of each solution category in addressing different categories of inverse problems.

The contributions in this paper focus on developing three categories of deep learning frameworks, applying each of these to five widespread, broadly-understood inverse problems, and then assessing the robustness in each case *via* statistical analysis. The specific contribution of this work is to develop a deeper understanding of the choice of best deep-learning framework for each type of inverse problem.

The rest of this paper is organized as follows: "Literature Review" includes a review of the most recent deep approaches to solving inverse problems; "Problem Definition" describes the problem definition, introducing three main categories for deep solutions for inverse problems; "Experiments" explains the experimental results including robustness analysis; finally "Conclusions" concludes the paper, proposing the best approach based on our experiments.

## LITERATURE REVIEW

Inverse problems have had a long history (*Engl, Hanke & Neubauer, 1996*; *Fieguth, 2010*; *Stuart, 2010*) in a wide variety of fields. In our context, since imaging involves the observing of a scene or phenomenon of interest, through a lens and spatial sensor, where the goal is to infer some aspect of the observed scene, essentially *all* imaging is an inverse problem, widely explored in the literature (*Bertero & Boccacci, 1998*; *Mousavi & Baraniuk, 2017*; *De los Reyes, Schönlieb & Valkonen, 2016*). Imaging-related inverse problems may fall under any of image recovery, restoration, deconvolution, pansharpening, concealment, inpainting, deblocking, demosaicking, super-resolution, reconstruction from projections, compressive sensing, and many others.

Inverse problems are ultimately the deducing of some function $G(\cdot)$ which *inverts* the forward problem in (1), with $\underline{z} = G(\underline{m})$, where some objective criterion obviously needs to be specified in order to select $G(\cdot)$. Since $G(\cdot)$ is very large (an input image has many pixels), unknown, and frequently nonlinear, it has become increasingly attractive to consider the role of DNNs, in their role as universal function approximators, in deducing $G(\cdot)$, and a number of approaches have been recently proposed in this fashion (*Lucas et al., 2018*; *Arridge et al., 2019*; *McCann & Unser, 2019*).

The most common approach when using DNNs for inverse problem solving includes optimizing the squared-error criterion $||\underline{z} - G(\underline{m})||_2^2$, with $G(\cdot)$ a DNN to be learned (*Adler & Öktem, 2017*; *Antholzer, Haltmeier & Schwab, 2019*; *Jin et al., 2017*; *Kelly, Matthews & Anastasio, 2017*; *Anirudh et al., 2018*; *Zhang & Ghanem, 2018*; *Fan et al., 2017*). This strategy implicitly finds a *direct mapping* from $\underline{m}$ to $\underline{z}$ using pairs $(\underline{z}, \underline{m})$ as the training data in the learning phase, which seeks to solve

$$\hat{W} = \arg_W \min ||\underline{z} - G(\underline{m}, W)||_2^2 \qquad (4)$$

for $W$ the network weights in the DNN, and $\underline{z}, \underline{m}$ as system parameters and measurements, respectively. Such supervised training needs a large number of data samples, which in some cases may be generated from the forward function $F(\cdot)$.

Recent work in direct mapping includes (*Häggström et al., 2019*), in which an encoder-decoder structure is proposed to directly solve clinical positron emission tomography (PET) image reconstruction. Similarly *Chen et al. (2019)* proposes a direct mapping deep

learning framework to identify the impact load conditions of shell structures based on their final state of damage, an inverse problem of engineering failure analysis.

Recent research investigates the incorporation of prior knowledge into DNN solutions for inverse problems. In particular, the use of intelligent initialization of DNN weights and analytical regularization techniques form the main classes of existing work in this domain (*Lucas et al., 2018*). In *Goh et al. (2019)*, variational autoencoders are used to solve forward and inverse problems, where the latent space of the autoencoder is used as the Parameter of Interest (PoI) space, and input and output of the autoencoder as the observation spaces. In *Anirudh et al. (2018)*, an unsupervised deep framework is proposed for solving inverse problems using a Generative Adversarial Network (GAN) to learn a prior without any information about the measurement process. In *Dittmer et al. (2019)*, a variational autoencoder (VAE) is used to solve electrical impedance tomography (EIT), a nonlinear ill-posed inverse problem. The VAE uses a variety of training data sets to generate a low dimensional manifold of approximate solutions, which allows the ill-posed problem to be converted to a well-posed one.

The forward model provides knowledge regarding data generation, based on the physics of the system. In *Chang et al. (2017)* an iterative variational framework is proposed to solve linear computer vision inverse problems of denoising, impainting, and super-resolution. It proposes a general regularizer $R$ for linear inverse problems which is first learned by a huge collection of images, and which is then incorporated into an Alternating Direction Method of Multipliers (ADMM) algorithm for optimizing

$$min_{\hat{\underline{z}}} \frac{1}{2}||\underline{m} - F\hat{\underline{z}}||_2^2 + \lambda R(\hat{\underline{z}}, W) \qquad (5)$$

Here regularizer $R(\cdot)$ was learned from image datasets and $W$ is the network weight matrix, as before. Here $F$ is a matrix, the (assumed to be) linear forward model.

The equivalent approach for a non-linear forward model is considered in *Li et al. (2018)*, in which a *data consistency* term $D(F(\hat{\underline{z}}), \underline{m})$ as a training objective incorporates the forward model into the problem:

$$min_{\hat{\underline{z}}} \{D(F(\hat{\underline{z}}), \underline{m}) + \lambda R(\hat{\underline{z}}, W)\} \qquad (6)$$

for regularization weight $\lambda$.

In *Senouf et al. (2019)*, a self-supervised deep learning framework is proposed for solving inverse problems in medical imaging using only the measurements and forward model in training the DNN.

Further DNN methods for inverse problems are explored in *Aggarwal, Mani & Jacob (2018)*, where the forward model is explicitly used in an iterative deep learning framework, requiring fewer parameters compared to direct mapping approaches. In *Yaman et al. (2019)*, an iterative deep learning framework is proposed for MRI image reconstruction. The work in *Bar & Sochen (2019)* proposes an unsupervised framework for solving forward and inverse problems in EIT. In *Cha, Lee & Oh (2019)* the analytical forward model is directly used in determining a DNN loss function, yielding an unsupervised framework utilizing knowledge about data generation. Other methods optimize data

consistency using an estimate of the forward model, learned from training data (*Fraccaro et al., 2017*).

A recent trend toward solving inverse problems involves estimating the posterior probability of the system parameters $p(\underline{z}|\underline{m})$ (*Dinh, Sohl-Dickstein & Bengio, 2016*; *Ardizzone et al., 2018*; *Kingma & Dhariwal, 2018*). An invertible structure is adopted to train the framework with the system parameters as input and measurements as network outputs. After training, the invertible network structure permits operating in the opposite direction, *i.e.*, accepting measurements as input and producing the desired estimates (*Ardizzone et al., 2018*). Our focus in this paper is on the objective function for categorization the solutions, therefore such invertible structures do not themselves introduce a separate solution category in our experiments.

The system parameters may themselves be coefficients in a partial differential equation (PDE) governing the system, whereby the observations are discrete measurements of the state variables of the PDE. Assuming that the observations are corrupted by additive noise, then function $F(\cdot)$ in (1) will be a PDE. In such Physics Informed inverse problems, specific deep learning structures have been developed, including Physics Informed Neural Networks (PINNs) (*Raissi, Perdikaris & Karniadakis, 2019*), in which the PDE is faced as a regularization term, or Quadratic residual (Qres) NNs (*Bu & Karpatne, 2021*) with greater expressive power. This class of inverse problems aims to find more complex solutions with less training data and achieving fewer parameters. In *Pakravan et al. (2021)* and *Goh et al. (2019)*, the parameters of an inverse problem in PDEs are considered as the latent space of an autoencoder, and are learned in an unsupervised manner. The work of *Goh et al. (2019)* uses a variational autoencoder, and *Pakravan et al. (2021)* aims to find the coefficients using a semantic autoencoder in which the decoder part is an analytic PDE solver.

The approach presented in *Maass (2019)* is closely related to ours, and aims at analysing deep learning structures for solving inverse problems, seeking to understand neural networks for solving small inverse problems. Our goal in this paper is to categorize deep learning frameworks for different inverse problems, based on their objectives and training schemes, investigating the power of each.

## PROBLEM DEFINITION

Recall the forward model (1):

$$\underline{m} = F(\underline{z}) + \underline{v} \qquad \underline{v} \sim N(0, I) \tag{7}$$

with given noise process $\underline{v}$, assumed to be white. There are three fundamental classes of inverse problems to solve:

- **Static Estimation Problems,** in which the system state $\underline{z}$ is static, without any evolution over time *Fieguth (2010)*. We will consider the following static problems:

– **Image Restoration,** part of a class of inverse problems in which the state and measurement spaces coincide (same number of pixels). Typically the measurements are a

corrupted version of the unknown state, and the problem is to recover an estimate of the true signal from its corrupted version knowing the (forward) distortion model.

– **Image Reconstruction,** to find a projection from some measurement space to a differently sized state, such as 3D shape reconstruction from 2D scenes. These problems need careful regularization to find feasible solutions.

- **Dynamic Estimation Problems,** in which $\underline{z}$ is subject to dynamics and measurements over time *Fieguth (2010)*, such as in object tracking.
- **Inverse Problems in Partial Differential Equations (PDEs)** refers to reconstruction of the parameters of a PDE, including coefficients, boundary conditions, initial conditions, the shape of domains, or singularity from partial knowledge of solutions to the PDE (*Raissi, Perdikaris & Karniadakis, 2019*; *Bu & Karpatne, 2021*; *Pakravan et al., 2021*; *Goh et al., 2019*).

Our focus is on DNNs as data-driven models for solving inverse problems, so we wish to redefine inverse problems to the context of learning from examples in statistical learning theory (*Vito et al., 2005*). We need two sets of variables:

$$\text{Inputs } \underline{m} \in M \qquad \text{Outputs } \underline{z} \in Z \tag{8}$$

The relation between input and output is described by a probability distribution $p(\underline{m}, \underline{z}) \in M \times Z$, where the distribution is known only through a finite set of samples, the training set

$$S = \{\underline{m}_i, \underline{z}_i\} \qquad 1 \leq i \leq N \tag{9}$$

assumed to have been drawn independently and identically distributed (i.i.d.) from $p$. The learning objective is to find a function $G(\underline{m})$ to be an appropriate approximation of output $\underline{z}$ in the case of a given input $\underline{m}$. That is,

$$\text{True } \underline{z} \approx \text{ Estimated } \underline{z} = G(\underline{m}|S), \tag{10}$$

such that $G(\cdot \mid S)$ was learned on the basis of $S$.

In order to measure the effectiveness of estimator function $G$ in inferring the desired relationship described by $p$, the expected conditional error can be used:

$$I(G) = \int_{M \times Z} D(G(\underline{m}), \underline{z}) \, dp(\underline{z}, \underline{m}) \tag{11}$$

where $D(G(\underline{m}), \underline{z})$ is the cost or *loss function*, measuring the cost associated with approximating true value $\underline{z}$ with an estimate $G(\underline{m})$. Choosing a squared loss $(G(\underline{m}) - \underline{z})^2$ allows us to derive

$$G(\underline{m}) = \int_Z \underline{z} \, dp(\underline{z}|\underline{m}) = E_p[\underline{z}|\underline{m}], \tag{12}$$

the classic optimal Bayesian least-squares estimator (*Fieguth, 2010*). In the case of learning from examples, (12) cannot be reconstructed exactly since only a finite set of

examples $S$ is given; therefore a regularized least squares algorithm may be used as an alternative (*Poggio & Girosi, 1989*; *Cucker & Smale, 2002*), where the hypothesis space $H$ is fixed and the estimate $G_S^\lambda$ is obtained as

$$G_S^\lambda = \arg_{G \in H} \min \left\{ \sum_{i=1}^{N} D(G(\underline{m}_i), \underline{z}_i) + \lambda R(G(\underline{m}_i)) \right\}, \tag{13}$$

where $R(\cdot)$ is a penalty term and $\lambda$ a regularization parameter which could be selected *via* cross-validation (*Arridge et al., 2019*).

Given that $H$ is the hypothesis space of possible inverse functions, in this paper it is quite reasonable to understand $H$ to be the space of functions which can be learned by a deep neural network, on the basis of optimizing its weight matrix $W$. Based on the optimization criterion (13), which is actually the variational framework in functional analytic regularization theory (*Poggio, Torre & Koch, 1985*), and which forms the basis for inverse-function DNN learning, we claim in this paper that, in terms of the objective criterion, each deep learning solution category lies in the one of the following three classes:

- Direct Mapping (DM)
- Data Consistency Optimizer (DC)
- Deep Regularizer (DR)

Each of these is developed and defined, as follows.

## Direct mapping

The direct mapping category is used as the objective criterion in a large body of research in deep learning based inverse problems (*Adler & Öktem, 2017*; *Antholzer, Haltmeier & Schwab, 2019*; *Jin et al., 2017*; *Kelly, Matthews & Anastasio, 2017*; *Anirudh et al., 2018*; *Zhang & Ghanem, 2018*; *Fan et al., 2017*). These methods seek to find end-to-end solutions for

$$\min_{W_1} \left\{ \sum_{i=1}^{N} D(\underline{z}, G(\underline{m}, W_1)) + \lambda R(G(\underline{m}, W_1)) \right\} \tag{14}$$

whereby $D(\cdot, \cdot)$ is the cost function to be minimized by a DNN $G(\underline{m}, W_1)$, on the basis of optimizing DNN weights $W_1$. $R(G(\underline{m}, W_1))$ specifies a generic analytical regularizer, to restrict the estimator to feasible solutions.

The Direct Mapping category approximates an estimator $G$ as an inverse to the forward model $F$, requiring a dataset of pairs $\{(\underline{m}_i, \underline{z}_i)\}_i$ of observed measurements and corresponding target system parameters, as illustrated in Fig. 1.

This category of DNN is typically used in those cases where we have a model-based imaging system having a linear forward model $\underline{m} = F(\underline{z})$, where $\underline{z}$ is an image, so that convolution networks (CNNs) are nearly always used. As discussed earlier, for Image Restoration problems the measurements themselves are already images, however in more general contexts we may choose to project the measurements as $F^H \underline{m}$, back into the domain of $\underline{z}$, such that the CNN is trained to learn the estimator

$$loss_{DM} = \left\| \underline{z} - G(\underline{m}, W_1) \right\|_2^2 + R(G(\underline{m}, W_1))$$

$\underline{m} \longrightarrow$ | **DNN** $G(., \boldsymbol{W_1})$ | $\longrightarrow G(\underline{m}, W_1)$

**Figure 1 Direct mapping of deep learning inverse problems.**

$$loss_{DC} = \left\| \underline{m} - F(G(\underline{m}, W_2)) \right\|_2^2 + R(G(\underline{m}, W_2))$$

$\underline{m} \longrightarrow$ **DNN** $G(., \boldsymbol{W_2})$ → Forward Model → $F(G(\underline{m}, W_2))$

**(A)**

$\underline{m} \longrightarrow$ **DNN** $G(., \boldsymbol{W_2})$ → $G(\underline{m}, W_2)$

**(B)**

**Figure 2 Data consistency optimization, where (A) the forward model is incorporated in the loss function of the DNN and is utilized during DNN training and (B) is removed in the inference time.**

$$\hat{\underline{z}} = G(F^H \underline{m}, W_1) \tag{15}$$

The translation invariance of $F^H F$, relatively common in imaging inverse problems, makes the convolutional-kernel nature of CNNs particularly suitable for serving as the estimator for these problems.

In general, the performance of direct inversion is remarkable (*Lucas et al., 2018*). However the receptive field (*i.e.*, the size of the field of view the unit has over its input layer) of the CNN should be matched to the support of the point spread function (*Aggarwal, Mani & Jacob, 2018*). Therefore, large CNNs with many parameters and accordingly extensive amount of training time and data are often needed for the methods in this category. These DNNs are highly problem dependent and for different forward models (*e.g.*, with different matrix sizes, resolutions, *etc.*) a new DNN will need to be learned.

### Data consistency optimizer

The Data Consistency Optimizer category of deep learning aims to optimize data consistency as an unsupervised criterion within a variational framework (*Aggarwal, Mani & Jacob, 2018*; *Cha, Lee & Oh, 2019*):

$$\min_{W_2} \left\{ \sum_{i=1}^{N} D(\underline{m}, F(G(\underline{m}, W_2))) + \lambda R(G(\underline{m}, W_2)) \right\} \tag{16}$$

where, as in (14), $D(\cdot, \cdot)$ is the cost function to be minimized by DNN $G(\underline{m}, W_2)$, parameterized by weights $W_2$, subject to regularizer $R(G(\underline{m}, W_1))$. The overall picture is summarized in Fig. 2.

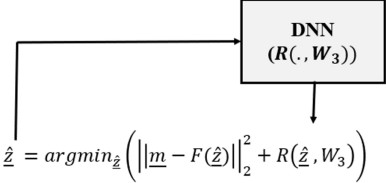

**Figure 3 Deep regularized category of inverse problems, in which a DNN is used only as the regularizer as part of an analytical variational framework.**

In contrast to (14), where the network cost function $D$ is expressed in the space of unknowns $\underline{z}$, here (16) expresses the cost in the space of *measurements* $\underline{m}$, based on forward model $F(\cdot)$. That is, the data consistency term is no longer learning from supervised examples, rather from the forward model we obtain an *unsupervised* data consistency term, not needing data labels, whereby the forward model provides some form of implicit supervision.

Compared to the direct mapping category, the use of the forward model in (16) leads to a network with relatively few parameters, in part because the receptive field of the DNN need not be matched to the support of the point spread function. However, the ill-posedness of the inverse problem causes a semi-convergent behaviour (*Arridge et al., 2019*) using this criterion, therefore an early stopping regularization needs to be adopted in the learning process.

## Deep regularizer

Finally, the Deep Regularizer (DR) category needs a different problem modeling scheme, since there is not a learning phase as in DM and DC. Instead, only a DNN (usually a classifier) is trained to be used as the regularizer in a variational optimization framework. That is, DR continues to optimize the data consistency term, however the overall optimization process is undertaken in the form of an analytical variational framework and uses a DNN as the regularizer (*Chang et al., 2017*; *Li et al., 2018*):

$$\min_{\hat{\underline{z}}} \left\{ \sum_{i=1}^{N} D(\underline{m}, F(\hat{\underline{z}})) + \lambda R(\hat{\underline{z}}, W_3) \right\} \tag{17}$$

Here $R(\hat{\underline{z}}, W_3)$ is a pre-trained deep regularizer, based on weight matrix $W_3$, usually chosen as a deep classifier (*Chang et al., 2017*; *Li et al., 2018*), discriminating the feasible solutions from non-feasible ones.

This category usually includes an analytical variational framework consisting of a data consistency term and a learned DNN to capture redundancy in the parameter space (see Fig. 3).

Since our interest is in the DNN solution of the inverse problem, and not the details of the optimization, we have chosen two fairly standard optimization approaches, a simplex/ Nelder-Mead approach *Singer & Nelder (2009)* (DR-NM) and a Genetic Algorithm strategy (DR-GA), both based on their respective Matlab implementations. Because GA

solutions may be different from one run to the next, in general we report the results averaged over multiple independent runs.

The Deep Regularizer category needs the fewest parameter settings, compared to the earlier categories; however because of the optimization based inference step it is computationally demanding.

# EXPERIMENTS

Our focus in this paper is to study solution robustness in the presence of noise and outliers during inference. This section explores experimental results, for *each* of the the fundamental inverse-problem classes (*restoration, reconstruction, dynamic estimation, physics informed*) for *each* of the categories of solution (*direct mapping (DM), data consistency optimizer (DC), deep regularizer (DR)*), as discussed in the previous section. Our study is based on a statistical analysis *via* the *Wilcoxon signed rank test* (*Lathuilière et al., 2019*), a well-known tool for analysing deep learning frameworks. The *null* hypothesis is that the result of each pairwise combination of DM, DC, and DR are from the same distribution, i.e., that the results are not significantly different. The experimental results are based on the following problems:

- Linear Regression: a *reconstruction problem*, with the aim of finding line parameters from the noisy/outlier sample points drawn from that line.
- Finding the coefficients of Burgers' PDE: an *Inverse problem in PDEs* as continuous time model (*Bu & Karpatne, 2021*), with the aim of finding PDE coefficients from a set of observed data.
- Image Denoising: a *restoration* problem, with the objective of recovering a clean image from noisy observations. We use both synthetic texture images and real images.
- Single View 3D Shape Inverse Rendering: a *reconstruction* problem, for which the domains of the measurements and system parameters are different. The measurements include a limited number of 2D points (input image landmarks) with the unknown state, to be recovered, a 3D Morphable Model (3DMM). We use a 3D model of the human face, based on eigen-faces obtained from principal component analysis.
- Single Object Tracking: a *dynamic estimation* problem, for which the goal is to predict the location (system parameter) of a moving object based on its (noisy) locations, measured in preceding frames. While this problem seems to belong to the class of restoration problems, the embedded state in this problem requires additional assumptions regarding the time-dynamics, and thus additional search strategies.

All DNNs were implemented using the KERAS library (*Chollet, 2015*) and ADAM optimizer (*Kingma & Ba, 2014*) on an NVIDIA GeForce GTX 1080 Ti. The DNN structures can be found in the corresponding subsection. All of the deep learning components in our experiments are trained for at most 100 epochs using the default learning rate in KERAS library. Table 1 summarizes the overall experimental setup for all problems.

**Table 1  The five inverse problems considered in our experiments.**

| Inverse problem | Measurements | Unknown parameters | Forward model | Training data |
|---|---|---|---|---|
| Linear Regression (Reconstruction) | 2D coordinates of $N$ drawn samples from the line | Slope, Intercept | Straight line plus noise | Synthetic: $\{(y_i, x_i)\}$ including Gaussian noise with heavy-tailed outliers |
| Burgers' PDE (PDE Inverse Problems) | Observations $\underline{m}$, provided by *Bu & Karpatne (2021)* | PDE parameters in Burgers': $\underline{m}_t + \lambda_1 \underline{mm}_x - \lambda_2 \underline{m}_{xx} = 0$ | Nonlinear PDE equation plus noise | Synthetic |
| Image Denoising (Restoration) | Noisy Image | Clean Image | Image plus noise | Synthetic: 5,000 gray scale texture images ($64 \times 64$) from stationary random process *Fieguth (2010)* including exponential number of pixel outliers with heavy tailed distribution |
| 3D Shape Rendering (Reconstruction) | Standard 2D landmarks on input face image | Parameters of a BFM 3D model | Noisy projection from 3D to 2D | Synthetic: 72 landmarks on 2D input image of a 3D human face generated by a Besel Face Model (BFM) *Aldrian & Smith (2012)* including 5% outliers in input 2D landmarks |
| Single Object Tracking (Dynamic Estimation) | Noisy location of a ball in a board from $n$ previous time steps to current step | True Location of the ball | True object locations plus noise | Synthetic: Sequences of a moving ball location with different random initial states and variable speeds including Gaussian noise for all measurements |

## Linear regression

We begin with an exceptionally simple inverse problem. Consider a set of one dimensional samples $\{(x^{(i)}, m_y^{(i)})\}_{i=1}^N$, subject to noise, with some number of the training data subject to more extreme outliers, as illustrated in Fig. 4.

As an inverse problem, we need to define the forward model, which for linear regression is simply

$$\underline{m}_y = \alpha \underline{x} + \beta + \underline{v}. \tag{18}$$

Since our interest is in assessing the robustness of the resulting inverse solver, the number and behaviour of outliers should be quite irregular, to make it challenging for a network to generalize from the training data. As a result, the noise $\underline{v}$ is random variance, plus heavy-tailed (power law) outliers, where the *number* of outliers is exponentially distributed.

For this inverse problem, the unknown state is comprised of the system parameters $\underline{z}^T = [\alpha, \beta]$. Thus linear regression leads to a reconstruction problem, for which the goal is to recover the line parameters from a sample set including noisy and outlier data points.

With the problem defined, we next need to formulate an approach for each of the three solution categories. For direct mapping (DM) and data consistency (DC), the training data and DNN structures are the same, shown in Fig. 5, where the DC approach includes an additional layer which applies the given forward model of (18). We used the KERAS library, in which a *Lambda* layer is designed for this forward operation.

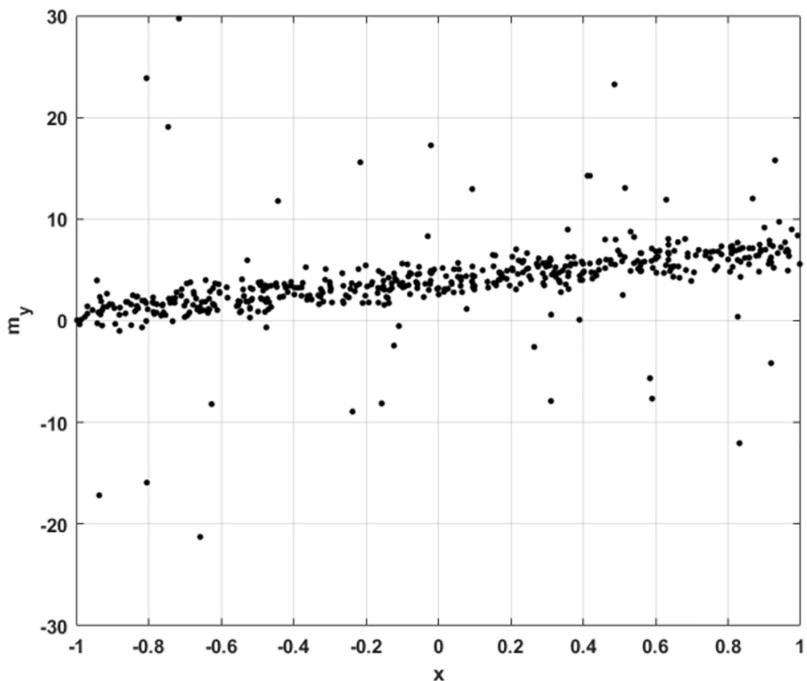

**Figure 4** **1D sample points for linear regression, with Gaussian noise and occasional large outliers.**

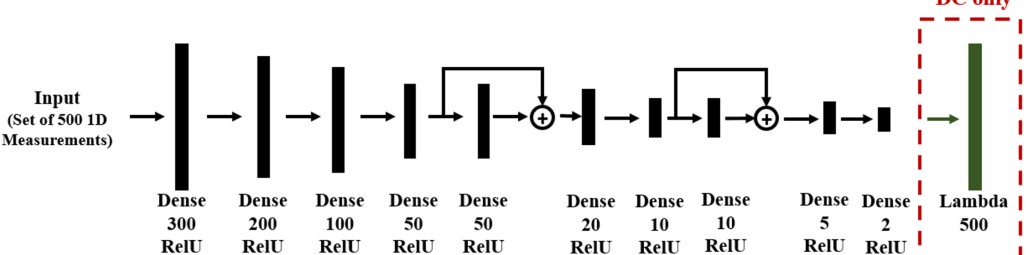

**Figure 5** **DNN structure for DM and DC solutions to linear regression. The layer type and number of neurons are reported below each layer.** Note that in the DC case, there is an additional *Lambda* layer, which computes the forward function from the predicted line parameters.

Since the problem is one-dimensional with limited spatial structure, the network contains only dense feed-forward layers. Residual blocks are used in order to allow gradient flow through the DNN and to improve training. Network training was based on 1,000 records, each of $N = 500$ noisy sample points.

The Deep Regularizer (DR) category needs a different problem modeling scheme, since 2there is not a learning phase as in DM and DC. Instead, only a DNN (usually a classifier) is trained to be used as the regularizer in a variational optimization framework. The DNN regularizer is given the system parameters $(\alpha, \beta)$ and determines whether they account for a feasible line. Here, we define the feasible line as a line having a tangent in some specified range. We generate a synthetic set of system parameters with associated

**Table 2 The error of estimated lines, with parameters averaged over 10 independent training/inference runs, obtained by the three DNN categories compared with least-squares.**

| Training data | Measure—Method | DM | DC | DR-GA | DR-NM ($z_0 = [0, 0]$) | LS |
|---|---|---|---|---|---|---|
| Noisy + Outlier | Error (Slope) | **0.23 ± 1.37** | 0.30 ± 1.27 | 0.96 ± 0.03 | 0.90 ± 0 | 0.61 ± 2.10 |
| | Error (Intercept) | 0.15 ± 1.68 | **0.06 ± 1.59** | 1.13 ± 0.04 | 1.09 ± 0 | 0.22 ± 3.00 |
| Noise-Free | Error (Slope) | 1.50 ± 2.08 | 1.26 ± 1.45 | 0.96 ± 0.03 | 0.90 ± 0 | **0.61 ± 2.09** |
| | Error (Intercept) | 0.32 ± 1.85 | 0.32 ± 1.38 | 1.13 ± 0.04 | 1.09 ± 0 | **0.21 ± 3.00** |

**Note:**
In each case, the best result is marked in bold.

**Table 3 Wilcoxon signed rank test $p$-values obtained for the linear regression problem, using noisy and outlier data for both training and testing.** We used 500 test samples to perform the statistical analysis over 10 independent training/inference steps of each method.

| $p$-value (Wilcoxon Test) | DM | DC | DR-GA | DR-NM | LS |
|---|---|---|---|---|---|
| DM | – | 0.695 | 0.002 | 0.002 | 0.002 |
| DC | 0.695 | – | 0.002 | 0.002 | 0.002 |
| DR-GA | 0.002 | 0.002 | – | 0.781 | 0.002 |
| DR-NM | 0.002 | 0.002 | 0.781 | – | 0.002 |
| LS | 0.002 | 0.002 | 0.002 | 0.002 | – |

labels for training a fully connected DNN as the regularizer for this category. Since our interest is in the DNN solution of the inverse problem, and not the details of the optimization, we have chosen two fairly standard optimization approaches, a simplex/Nelder-Mead approach *Singer & Nelder (2009)* and a Genetic Algorithm (GA) strategy, both based on their respective Matlab implementations. Because GA solutions may be different over multiple runs, we report the results averaged over ten independent runs.

Table 2 shows the average solution found by each category over 10 independent trainings for DM and DC, and 10 independent inferences for DR. The table also reports *Least-Squares (LS)* results as a point of reference method, particularly to show the improvement that deep learning methods have to offer for robustness in solving inverse problems. Observe the significant difference when the DNN methods are trained with noise-free as opposed to noisy data, such that the noisy training data force the network to acquire a robustness to outliers.

For DR we trained a 5 layer MLP with dense layers of sizes 5, 4, 3, 2, 1, as the regularizer, using the generated synthetic data including feasible line parameters (in the specific range) as the positive training samples and invalid line parameters as the negative training samples. The average test accuracy of the trained regularizer is 95.70%.

We performed the *Wilcoxon signed rank test*, for both cases of training with noisy data (Table 3) and noise-free training (Table 4). The tables show the pairwise p-values over the 10 independent runs. A *p-value* in excess of 0.05 implies that the two methods are likely to stem from the same distribution; in particular, the Wilcoxon test computes the probability that the difference between the results of two methods are from a distribution with median equal to zero. Clearly all of the DNN methods are statistically significantly

**Table 4  Like Table 3, but now using noise-free data, *i.e.*, without any noise or outliers, for method training.** Noisy and outlier data remain in place for testing.

| p-value (Wilcoxon Test) | DM | DC | DR-GA | DR-NM | LS |
|---|---|---|---|---|---|
| DM | – | 0.002 | 0.002 | 0.002 | 0.002 |
| DC | 0.002 | – | 0.002 | 0.002 | 0.002 |
| DR-GA | 0.002 | 0.002 | – | 0.781 | 0.002 |
| DR-NM | 0.002 | 0.002 | 0.781 | – | 0.002 |
| LS | 0.002 | 0.002 | 0.002 | 0.002 | – |

different from the least-squares (LS) results. For noisy training data, the statistical results in Table 3 show similar performance for DM and DC, and for DR-NM and DR-GA, the latter similarity suggesting that the specific choice of optimization methodology does not significantly affect the DR performance.

The results in Table 2 show that DM and DC significantly improve in robustness when trained with noisy data, relative to training with noise-free data. The principal difference between DM/DC *vs* DR is the learning phase for DM/DC, allowing us to conclude that, at least for reconstruction problems, a learning phase using noisy samples in training significantly improves the robustness of the solution. A further observation is that whereas DM and DC achieve similar performance, DC is unsupervised and DM is supervised. Thus it would appear that the forward model knowledge and the data consistency term as objective criterion for DC provide an equal degree of robustness compared to the supervised learning in DM.

For this reconstruction problem, we conclude that both DC and DM perform well, with the unsupervised DC showing strong performance both with noisy and noise-free training data.

## Finding coefficients of Burgers' PDE (inverse problems in PDEs)

To test deep learning for Inverse Problems in PDEs, we chose *Burgers'* Partial Differential Equations (PDEs) as a dynamic, continuous time PDE in our experiments.

Burgers' PDE or Bateman–Burgers equation (*Bateman, 1915*; *Burgers, 1948*) is a basic partial differential equation occurring in various areas of applied mathematics, such as fluid mechanics, nonlinear acoustics, gas dynamics, and traffic flow *Xin (2009)*. This setup encapsulates a wide range of problems in the mathematical physics including conservation laws, diffusion processes, advection-diffusion-reaction systems, and kinetic equations (*Raissi, Perdikaris & Karniadakis, 2019*).

For a given field $\underline{m}(x, t)$, we consider Burgers' equation as the forward function, defined as

$$F(\underline{m}(x, t); \underline{z}) = \underline{m}_t + z_1 \underline{m}\,\underline{m}_x - z_2 \underline{m}_{xx} = 0 \quad x \in \Omega, \quad t \in [0, T] \tag{19}$$

where $z_1, z_2$ denote the parameters of the equation and $\underline{m}(x, t)$ the state of the system, the subscripts denoting partial differentiation in either time or space. The goal is then to estimate parameters $z_1, z_2$, given a collection of points (*Raissi, Perdikaris & Karniadakis, 2019*; *Bu & Karpatne, 2021*).

Solutions for this type of inverse problems in the literature, including Physics Informed Neural Networks (PINNs) (*Raissi, Perdikaris & Karniadakis, 2019*), and Quadratic Residual Neural Networks (QRes) (*Bu & Karpatne, 2021*) actually use a regularized version of data loss, with $F(\underline{m};\underline{z})^2$ as the regularizer. For instance, PINNs are defined as

$$G := F(\underline{m}(x,t);\underline{z}) \tag{20}$$

with $F$ from (19), and using a DNN to approximate $\underline{m}(x,t)$. The DNN, along with (20), form the *Physics Informed Neural Network* $G(t,x)$ in which the chain rule could be used for differentiating compositions of functions using automatic differentiation (*Baydin et al., 2018*), which we call *AutoGrad*, and has the same parameters as the network representing $G(t,x;W_2)$, albeit with different activation functions due to the action of $F$. The shared parameters between the neural networks $m(t,x)$ and $G(t,x;W_2)$ can be learned by minimizing the mean squared error loss

$$J_{DM} = PGLoss + DataLoss \tag{21}$$

where

$$DataLoss = \frac{1}{N_m} \sum_{i=1}^{N_m} |\underline{m}(t_m^i, x_m^i; W_1) - \underline{m}^i|^2 \tag{22}$$

and

$$PGLoss = \frac{1}{N_G} \sum_{i=1}^{N_G} |G(t_G^i, x_G^i; W_2)|^2 \tag{23}$$

where $\{t_m^i, x_m^i, \underline{m}^i\}$ denote the initial and boundary training data on $\underline{m}(t,x)$ and $\{t_G^i, x_G^i\}_{i=1}^{N_G}$ specify the collocations points for $G(t,x)$. DataLoss corresponds to the initial and boundary data while PGLoss enforces the structure imposed by (19) at a finite set of collocation points.

This definition of the loss functions makes them consistent with the objective of the *Data Consistency optimizer (DC)* solution category. Therefore, we include these methods within DC in our experiments.

In the case of the DM approach, we can define the loss function as

$$J_{DM}(\underline{m}(x,t);\underline{z}) = \frac{1}{N_G} \sum_{i=1}^{N_G} |z_{pred}^i - z^{GT}|^2 + |G(t_G^i, x_G^i; W_2)|^2 \tag{24}$$

where $z_{pred}^i$, $z^{GT}$ stand for the predicted parameter by the solution category and its ground truth, respectively.

Figure 6 shows the DM and DC DNNs for Burgers' Inverse problem, where *AutoGrad*, the automatic differentiation component, is used for computing the needed gradients. The DC solution category only uses the PGLoss in its training procedure.

For DR, we define the loss function as

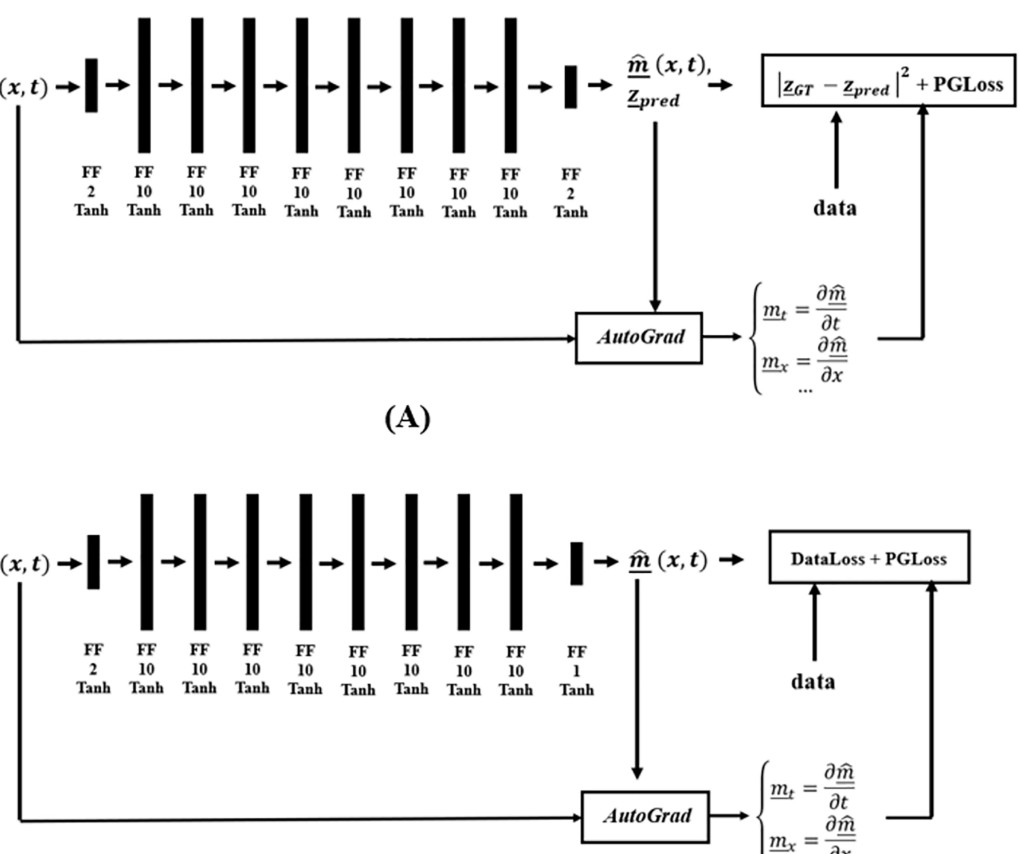

**Figure 6** Network structure for the (A) DM and (B) DC solution categories, in finding coefficients of Burgers' PDE.

$$J_{DR} = \frac{1}{N_G} \sum_{i=1}^{N_G} |G(t_G^i, x_G^i)|^2 + \lambda R(\underline{m}; W_3) \tag{25}$$

where $\frac{1}{N_G} \sum_{i=1}^{N_G} |G(t_G^i, x_G^i)|^2$ is the data consistency term and $R(\underline{m}; W_3)$ is a deep classifier, for which we trained an MLP classifier with dense layers of size 5, 4, 3, 2, 1, trained by he available measurement states, to control the values of $m$ to be in the specified range, provided by *Bu & Karpatne (2021)*.

For the experiments, we used synthetic data provided by *Bu & Karpatne (2021)* as the training and test data, where the standard deviation of the noise is set to 1% of the data standard deviation, and for $x_m^i$, $t_m^i$ we used equi-spaced values in the specified ranges. In the case of outliers, we used additive Gaussian noise with magnitude equal to 10 times the data standard deviation for 0.05% of the data. Table 5 compares the MSE between the obtained parameter values by existing methods, averaged over 5 independent training/inferences.

The statistical analyses of the results in Table 5 are reported in Table 6. From Tables 5 and 6, it is observable that in this case it is DC which achieves the best robustness

**Table 5 MSE of the parameters of Burgers' PDE, predicted by deep learning solution categories for this problem.**

| Training data | Measure—Method | DM | DC | DR-GA | DR-NM ($z_0 = [0, 0]$) |
|---|---|---|---|---|---|
| Noise-Free | Error ($z\_1$) ($\times 10^{-4}$) | **0.1 ± 0.10** | 2.7 ± 0.12 | 5.3 ± 3.5 | 5.1 ± 0 |
| | Error($z\_2$) ($\times 10^{-4}$) | 131.4 ± 51.0 | **37.9 ± 3.3** | 117.13 ± 83.5 | 100.1 ± 0 |
| Noisy | Error ($z\_1$) ($\times 10^{-4}$) | **0.3 ± 0.1** | 17.0 ± 9.2 | 5.3 ± 3.5 | 5.1 ± 0 |
| | Error($z\_2$) ($\times 10^{-4}$) | 42.1 ± 3.3 | **3.1 ± 0.8** | 117.13 ± 83.5 | 100.1 ± 0 |
| Noisy + Outlier | Error ($z\_1$) ($\times 10^{-4}$) | 31.2 ± 11.0 | 2.7 ± 13.0 | 0.53 ± 3.5 | 5.1 ± 0 |
| | Error($z\_2$) ($\times 10^{-4}$) | 67.0 ± 32.0 | **0.60 ± 32.0** | 117.13 ± 83.5 | 100.10 ± 0 |

Note:
   In each case, the best result is marked in bold.

**Table 6 Statistical analysis of the results in Table 5, using Wilcoxon signed rank test, in the case of all types of measurements, including noise-free, noisy and noisy and outlier case, for training the networks.**

| $p$-value (Wilcoxon Test) | DM | DC | DR-GA | DR-NM |
|---|---|---|---|---|
| DM | – | 0.005 | 0.005 | 0.005 |
| DC | 0.005 | – | 0.005 | 0.005 |
| DR-GA | 0.005 | 0.005 | – | 0.834 |
| DR-NM | 0.002 | 0.002 | 0.834 | – |

performance. The statistical analysis shows that the choice of DR optimization method does not impact the results. The results also show that the learning phase in DC significantly improves the obtained results compared with DR under the same objective.

## Image denoising (restoration)

We now consider an image denoising problem, following the steps described in "Linear Regression" for regression. We consider real and synthetic images, including 5 classes and 1,200 training images, 400 test images per class, from the **Linnaeus** dataset (*Chaladze & Kalatozishvili, 2017*) as real data, and synthesized 5,000 texture images generated by sampling from stationary periodic kernels, as synthetic data.

The synthetic images are generated using an FFT method (*Fieguth, 2010*), based on a thin-plate second-order Gauss–Markov random field kernel

$$
\mathscr{P} = \begin{bmatrix} 0 & 0 & 1 & 0 & 0 \\ 0 & 2 & -8 & 2 & 0 \\ 1 & -8 & 20 + \alpha^2 & -8 & 1 \\ 0 & 2 & -8 & 2 & 0 \\ 0 & 0 & 1 & 0 & 0 \end{bmatrix} \tag{26}
$$

such that a texture $T$ is found by inverting the kernel in the frequency domain,

$$
T = FFT_2^{-1}\left( \sqrt{1 \oslash FFT_2(\mathscr{P})} \odot FFT_2(W) \right), \tag{27}
$$

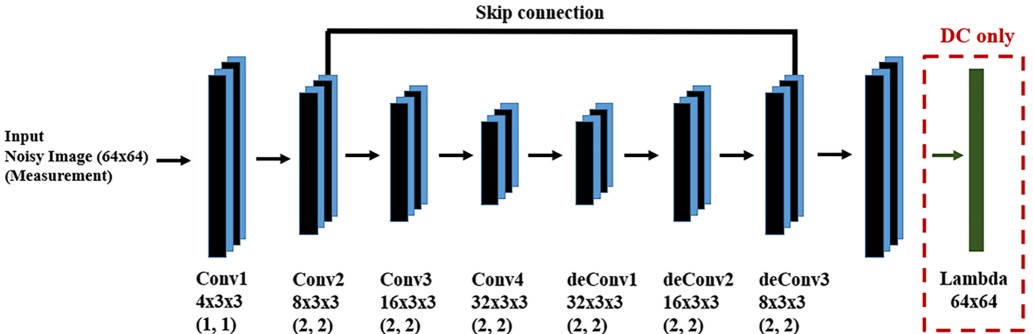

**Figure 7 DNN for the DM and DC solutions for image denoising problem.** We have a fully convolutional DNN with an encoder-decoder structure, where the values in parentheses indicate the stride value of the corresponding convolutional layer. The skip connection helps to recover desirable information which may be lost during encoding.

with $\odot$, $\oslash$ as element-by-element multiplication and division, $W$ as unit-variance white noise, and with the kernel $P$ zero-padded to the intended size of $T$. Further details about this approach can be found in *Fieguth (2010)*.

Parameter $\alpha^2$, affecting the central element of the kernel $P$, effectively determines the texture spatial correlation-length in $T$, as

$$\alpha^2 = 10^{4 - log_{10} u} \tag{28}$$

for process correlation length, $u$, measured in pixels. We set $u$ to be a random integer in the range [10, 200] in our experiments.

All images are set to be $64 \times 64$ in size, with pixel values normalized to [0, 1]. Pixels are corrupted by additive Gaussian noise, with an exponentially distributed number of outliers.

The inverse problem is a restoration problem, having the objective of restoring the original image from its noisy/outlier observation. The linear forward model is

$$\underline{m} = \underline{z} + \underline{v} \tag{29}$$

for measured, original, and added noise, respectively. The Gaussian noise $v$ has zero mean and random variance, and an exponential number of pixels become outliers, their values replaced with a uniformly distributed random intensity value.

We used 5,000 training samples and 500 test samples for the learning and evaluation phases of the DM and DC approaches. The DNN structure for both DM and DC is the same and is shown in Fig. 7. In the case of DC, we design a DNN layer to compute the forward function. Since we are dealing with input images, both as measurements and system state, we design a fully convolutional DNN in an encoder-decoder structure, finding the main structures in the image through encoding and recovering the image *via* decoding. Since there may be information loss during encoding, we introduce skip connections to help preserve desirable information.

The DR category needs a pre-trained regularizer which determines whether the prediction is a feasible texture image. We trained a classifier for texture discrimination,

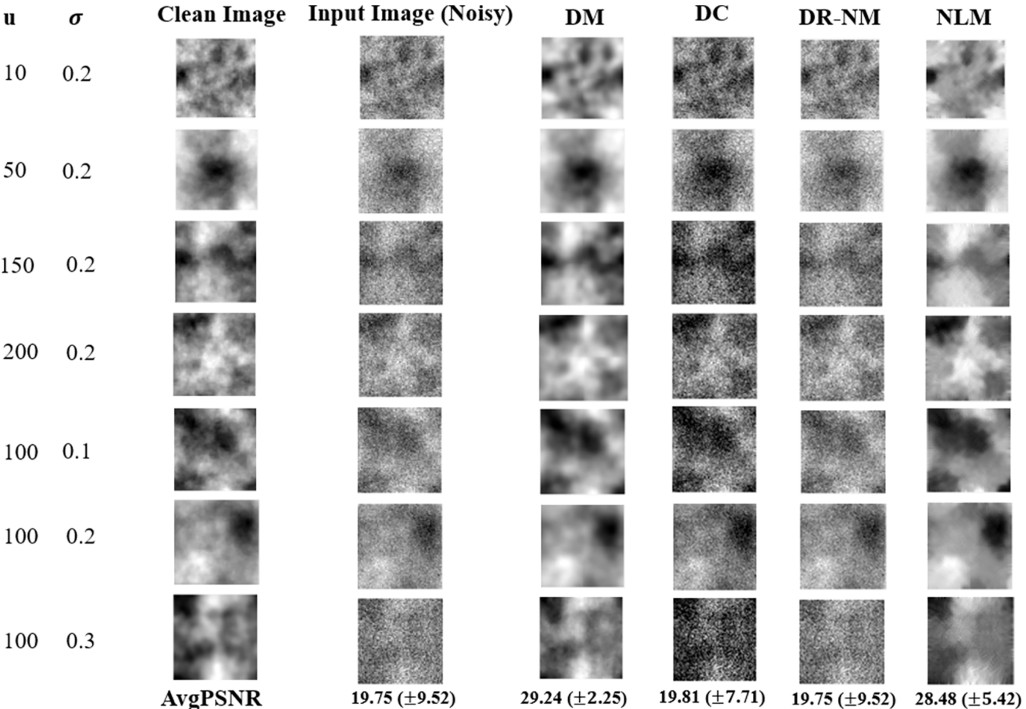

| u | σ | Clean Image | Input Image (Noisy) | DM | DC | DR-NM | NLM |
|---|---|---|---|---|---|---|---|
| 10 | 0.2 | | | | | | |
| 50 | 0.2 | | | | | | |
| 150 | 0.2 | | | | | | |
| 200 | 0.2 | | | | | | |
| 100 | 0.1 | | | | | | |
| 100 | 0.2 | | | | | | |
| 100 | 0.3 | | | | | | |
| | | **AvgPSNR** | 19.75 (±9.52) | 29.24 (±2.25) | 19.81 (±7.71) | 19.75 (±9.52) | 28.48 (±5.42) |

**Figure 8** **Image denoinsing results on synthetic textures.** Only a single image is shown in each case, however the reported average PSNR at the bottom is computed over the entire test set. The given noisy image is subject to both additive noise and outliers. NLM, in the rightmost column, is the non-local means filter, a standard approach from image processing.

generated using (27), from ordinary images gathered from the web, as the regularizer. Both GA and Nelder-Mead optimizers are used.

We use peak signal to noise ratio (PSNR) as the evaluation criterion, computed as

$$PSNR(I^{pred}, I^{GT}) = 20 \cdot log_{10} \max(I^{pred}) - 10 \cdot log_{10} MSE, \qquad (30)$$

$$MSE = \frac{1}{n} \sum_{i,j} (I^{GT}_{i,j} - I^{pred}_{i,j})^2 \qquad (31)$$

where $I^{GT}_{i,j}$, $I^{pred}_{i,j}$ are the $(i,j)^{th}$ pixel in the ground-truth and predicted images, respectively. Note that in the DR case, since the input and output of the model are 64 * 64 = 4,096 images, the GA optimization routine was unable to find the solution in a reasonable time, therefore we do not avoid report any DR-GA results for this problem.

As a reference point, we also report results obtained by the non-local means (NLM) filter (*Buades, Coll & Morel, 2011*), to give insight into the amount of improvement of deep learning inverse methods over a well-established standard in image denoising.

Figure 8 shows results based on synthetic textures. Each row in the figure shows a sample image associated with a particular correlation length noise standard deviation. The DM approach offers by far the best reconstruction among the DNN methods, and outperforms NLM in terms of PSNR. The time complexity of GA in DR-GA makes it inapplicable to problems of significant size (even though the images were still quite modest in size).

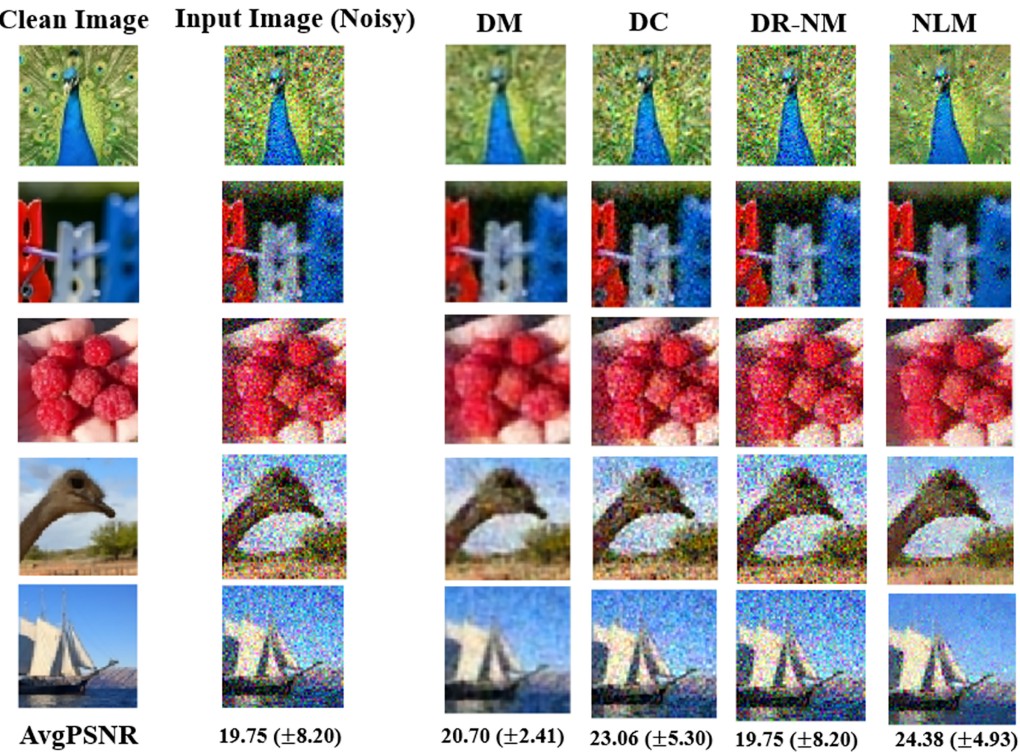

**Figure 9** **As in** **Fig. 8,** **but here for denoising results on the Linnaeus dataset.** The reported average PSNR in the last row is computed over all test images. As in Fig. 8, the DM results other DNN inverse solvers and also non-local means (NLM).

The Wilcoxon signed rank test was performed on the DM, DC and DR-(Nelder-Mead) results. The statistical analysis of the obtained results gave a $p$ value of 0.002 for each pairwise comparison, implying a statistically significant difference, thus the very strong performance of DM in Fig. 8 is validated.

In the case of real images, Fig. 9 shows the visual results obtained by DM, DC and DR-NM for seven test samples.

The statistical analysis is consistent with the results from the synthetic texture case, which is that all pairwise Wilcoxon tests led to a conclusion of statistically significant differences, with $p$ values well below 0.05.

From the results in Figs. 8 and 9 and their respective statistical analyses, we conclude that:

- For image denoising as a prototype for restoration problems, which have the same measurement and system parameter spaces, the concentration of the loss function on the true parameters (as in DM) provides better information and leads to a more effective estimator having greater robustness than the measurements themselves (as in DC).
- DR-(Nelder-Mead) performed poorly, even though it optimizes data consistency, like DC, however we believe that the learning phase in DC, compared to DR, provides knowledge for its inference and allows DC to be more robust than DR for restoration inverse problems.

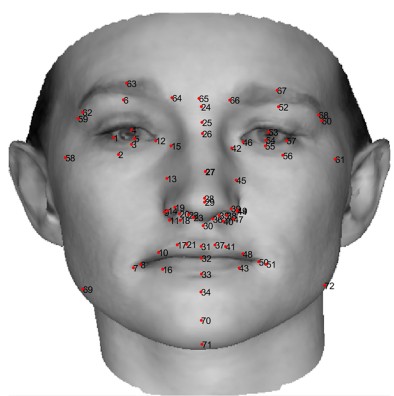

**Figure 10 Location and order of 72 standard landmarks on a 2D image of a sample human face.** The human face image in this figure is generated using BFM by *Blanz & Vetter (1999)* © Shima Kamyab.

### 3D shape inverse rendering (reconstruction)

We now wish to test a 3D shape inverse rendering (IR) (*Aldrian & Smith, 2012*) problem, for which a 3D morphable model (3DMM) (*Blanz & Vetter, 1999*) describes the 3D shape of a human face $\underline{s}$. This model is based on extracting eigenfaces $\underline{s}_i$, usually using PCA, from a set of 3D face shapes as the training data, then to obtain new faces as a weighted combination $z_i$ of the eigenfaces. The 3D shape model reconstructs a 3D face in homogeneous coordinates as

$$\underline{s} = \bar{\underline{s}} + \sum_{i=1}^{n} z_i \underline{s}_i, \tag{32}$$

where $\bar{\underline{s}}$ is the mean shape of the 3DMM, and $z_i$ the weight of eigenface $\underline{s}_i$. We use the Besel Face Model (*Aldrian & Smith, 2012*) as the 3DMM in this experiment for which there are $N = 54{,}390$ 3D points in each face shape and 199 eigenfaces. We can therefore rewrite (32) as

$$\underline{s}_N = \bar{\underline{s}}_N + \underline{z}^T * S_N \tag{33}$$

where $S$ is the tensor of 199 eigenfaces. In our experiments each face is characterized by 72 standard landmarks, shown in Fig. 10, which are normalized and then presented to the system as the measurements. Therefore we actually only care about $L = 72$ out of $N = 54{,}390$ 3D points in the 3DMM. This experiment tackles the reconstruction of a 3D human face by finding the weights $\underline{z}$ of the 3DMM from its input 2D landmarks. We generated training data from the 3DMM by assigning random values to the 3DMM weights, resulting in a 3D human face, and rendered the obtained 3D shape into a 2D image using orthographic projection.

The measurement noise consists of small perturbations of the 2D landmarks, with outliers as much larger landmark perturbations. We add zero-mean Gaussian noise having a standard deviation of $3 \times 10^3$ in the training data and $5 \times 10^3$ in the test data. Outliers are much larger, with a standard deviation of $5 \times 10^4$ added to 10 of the 72 landmarks

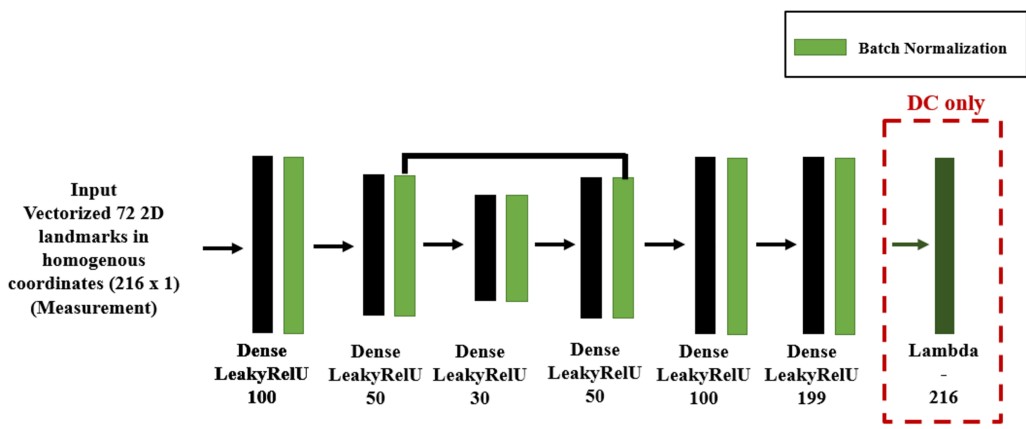

**Figure 11 DNN structure for DM and DC for 3D shape inverse rendering.**

in 10% of the training data and 20% of the test data. Landmark point coordinates are in the range $[-8 \times 10^4, 8 \times 10^4]$, so the outlier magnitudes are very large.

Let subscript $L$ represent the the set of landmark point indices, in which case the forward model is the orthographic projection

$$\underline{m} = C\underline{s}_L + \underline{v} \qquad C = \begin{bmatrix} 1 & 0 & 0 & 0 \\ 0 & 1 & 0 & 0 \\ 0 & 0 & 0 & 0 \end{bmatrix} \tag{34}$$

such that $C$ converts from homogeneous 3D to homogeneous 2D coordinates, and the measurement noise is

$$\underline{v} \sim 0.9N(0, 3 \times 10^3 I) + 0.1N(0, 5 \times 10^4 I) \tag{35}$$

as noise and outliers associated with the projection operator. Since the goal of this inverse problem is to estimate $\underline{z}$ in the 3DMM for a given 3D shape, we write (34) as

$$\underline{m} = C(\bar{\underline{s}}_L + S_L * \underline{z}) + \underline{v} \tag{36}$$

For the DM and DC solutions we generated 4,000 sample faces as training data, using the Besel face model (*Aldrian & Smith, 2012*) as the 3DMM. The DR regularizer is a pre-trained classifier which discriminates a feasible 3D shape from random distorted versions of it.

In DC we implemented the forward function layer as described in *Aldrian & Smith (2012)*, with the resulting DM and DC DNN shown in Fig. 11, where we used feed-forward layers because the system input is the vectorized 72 2D homogeneous coordinates and its output a weight vector. We design an encoder-decoder structure for DNNs, so as to map the 2D coordinates to a low dimensional space and to recover the parameters from that low dimensional representation.

For the DR regularizer we trained a five layer MLP classifier to discriminate between a 3D face shape, generated by BFM, and randomly generated 3D point clouds as negative examples.

Figure 12 shows visual results obtained by each solution category, where heat maps visualize the point-wise error magnitude relative to the ground truth. The visual results show that the DM and DC methods can capture the main features in the face (including eye, nose, mouth) better than the DR variants, however the differences between DM and DC seem to be negligible.

To validate our observations, the numerical results and respective statistical analyses are shown in Tables 7 and 8. Table 7 lists the RMSE values for each solution category. We used 10 out of sample faces in the BFM model as test cases for reporting the results. In the case of DR (Nelder-Mead) we set the start point, *i.e.*, $z_0$, as a random value and report the averaged result over 10 independent runs. Note that the RMSE values are expected to be relatively large, since each 3D face shape provided by BFM is a point cloud of 53,490 3D coordinates in the range $[-8 \times 10^4, 8 \times 10^4]$. As a point of comparison, we computed the average RMSE between a set of 500 generated 3D faces and 1,000 random generated faces, to have a sense of RMSE normalization to random prediction. The average RMSE for random prediction is $1.28 \times 10^4$, a factor of two to four times larger than the RMSE values reported in Table 7.

Table 8 shows the results of the Wilcoxon $p$ values for statistical significance in the difference between reported values in Table 7, where we consider a $p$ value threshold of 0.07.

Based on the preceding numerical results and statistical analysis, we claim the following about each solution category facing with **Reconstruction** inverse problems:

- Broadly, for training and test data not involving outliers, the overall performance of the methods is similar, with DM outperforming. This observation shows that the learning phase is not crucial in the presence of noise, and methods which concentrate on the test data can achieve equal performance compared to trainable frameworks.
- In cases involving outliers the performance of the methods is more distinct, but with the DM and DC methods, having a learning phase for optimizing their main objective term, outperforming the DR variants. We conclude that a learning phase is important to make methods robust to outliers.
- In the case of DR, the results show similar performance of the GA and NM optimization schemes, with GA outperforming NM. This observation encourages the reader to use optimization methods with more exploration power *Eftimov & Korošec (2019)*, the ability of an optimization method to search broadly across the whole solution space, for DR solutions to reconstruction problems.
- In all cases, we can observe that although DC is unsupervised, its performance when solving reconstruction inverse problems is near to that of DM, even outperforming DM in the case of outliers. Therefore, it is possible to solve reconstruction problems even without label information in the training phase.
- One interesting observation is that while 3D shape inverse rendering is a complex reconstruction problem, the results for each solution category are qualitatively similar to

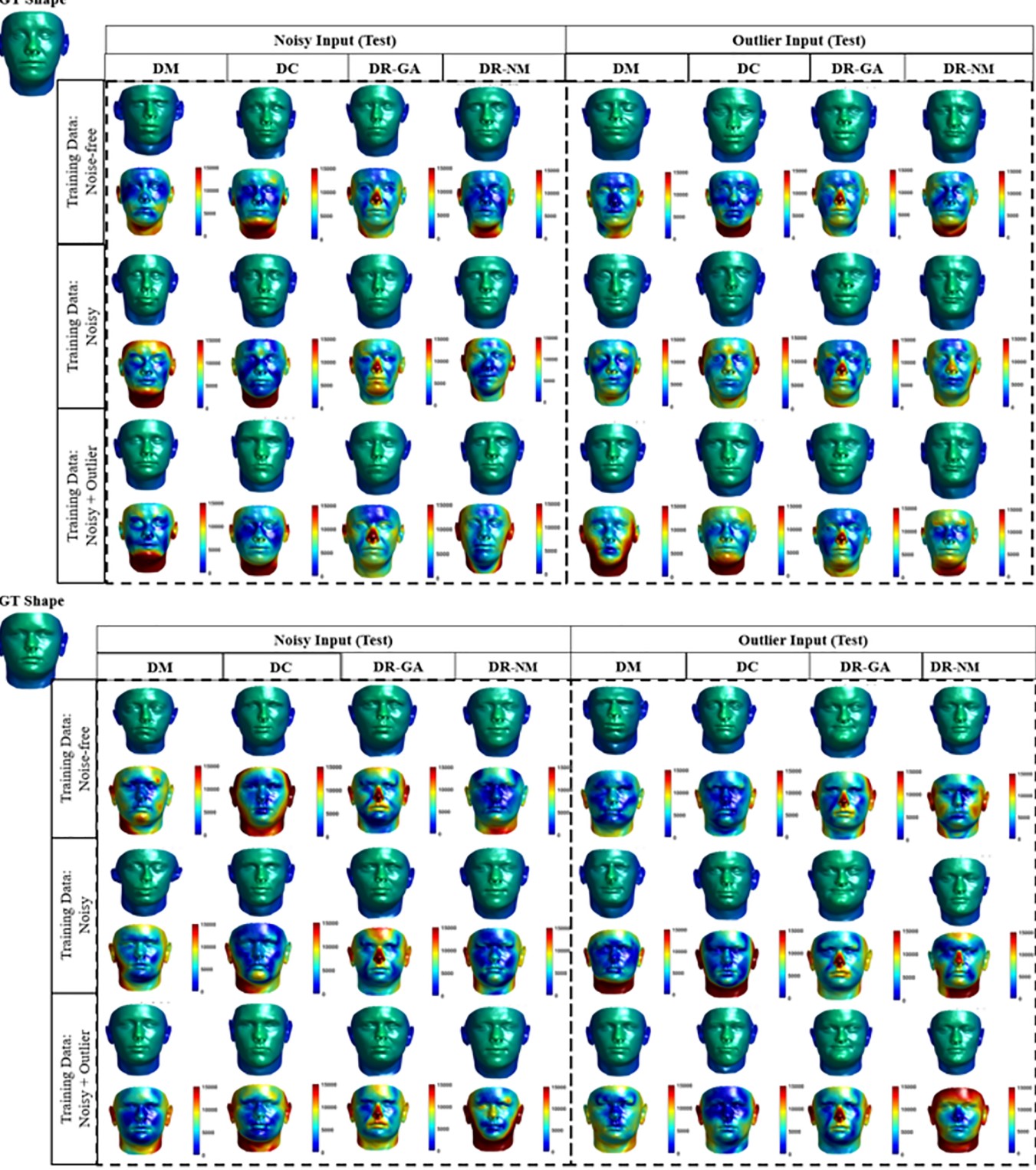

**Figure 12 Qualitative results for 3D inverse rendering.** Each result is shown as two faces, an upper with the actual 3D result, and a lower as a heat map showing the error magnitude in each point of predicted face are shown in the form of heat map for each prediction. For the DR method, the average error magnitude over 20 runs is reported. We use the Besel Face Model (BFM) (*Aldrian & Smith, 2012*; *Blanz & Vetter, 1999*) which is based on a 3D mean face and compensates for outliers. The BFM data was provided by Dr T. Vetter, University of Besel (*Gerig et al., 2018*).

**Table 7 Average test RMSE with standard deviation values (over 10 out-of-sample faces of the BFM (*Aldrian & Smith, 2012*)) for 3D shape inverse rendering.**

| Method | Noisy test cases ($\times 10^3$) | | | | Noisy + Outlier test cases ($\times 10^3$) | | | |
|---|---|---|---|---|---|---|---|---|
| Training data | DM | DC | DR-GA | DR-NM | DM | DC | DR-GA | DR-NM |
| Noise-free | **3.8 ± 2.0** | 4.2 ± 1.8 | 3.9 ± 0.7 | 4.2 ± 2.2 | 5.9± 2.4 | 5.5 ± 2.2 | 5.7 ± 1.2 | 5.8 ± 3.3 |
| Noisy | **3.5 ± 1.6** | 4.2 ± 2.1 | 3.9 ± 0.7 | 4.2± 2.2 | **5.4 ± 3.5** | 5.7 ± 3.6 | 5.7± 1.2 | 5.8 ± 3.3 |
| Noisy + Outlier | **3.3 ± 1.4** | 3.9 ± 1.8 | 3.9 ± 0.7 | 4.2 ± 2.2 | **5.4 ± 2.9** | 5.4 ± 3.0 | 5.7± 1.2 | 5.8 ± 3.3 |

Note:
In each case, the best result is marked as bold.

**Table 8 Wilcoxon signed rank test *p* values for the 3D shape inverse rendering problem.**

| Test data | Noisy | | | | | Noisy + Outlier | | | |
|---|---|---|---|---|---|---|---|---|---|
| Training data | *p*-value | DM | DC | DR-GA | DR-NM | DM | DC | DR-GA | DR-NM |
| Noise-free | DM | – | 0.19 | 0.43 | 0.30 | – | 0.06 | 0.06 | 0.06 |
| | DC | 0.19 | – | 0.43 | 0.30 | 0.06 | – | 0.06 | 0.06 |
| | DR-GA | 0.43 | 0.43 | – | 0.78 | 0.06 | 0.06 | – | 0.78 |
| | DR-NM | 0.30 | 0.30 | 0.78 | – | 0.06 | 0.06 | 0.78 | – |
| Noisy | DM | – | 0.19 | 0.19 | 0.30 | – | 0.06 | 0.06 | 0.06 |
| | DC | 0.19 | – | 0.30 | 0.30 | 0.06 | – | 0.12 | 0.30 |
| | DR-GA | 0.19 | 0.30 | – | 0.78 | 0.06 | 0.12 | – | 0.78 |
| | DR-NM | 0.30 | 0.30 | 0.78 | – | 0.06 | 0.30 | 0.78 | – |
| Noisy + Outlier | DM | – | 0.06 | 0.06 | 0.06 | – | 1 | 0.06 | 0.06 |
| | DC | 0.06 | – | 0.06 | 0.06 | 1 | – | 0.06 | 0.06 |
| | DR-GA | 0.06 | 0.06 | – | 0.78 | 0.06 | 0.06 | – | 0.78 |
| | DR-NM | 0.06 | 0.06 | 0.78 | – | 0.06 | 0.06 | 0.78 | – |

the very different and far simpler inverse problem of linear regression, where DC similarly outperformed training data containing noisy and outlier samples.

## Single object tracking (dynamic estimation)

Up to this point we have investigated deep learning approaches applied to static problems. We would now like to examine a dynamic inverse problem, that of single-object tracking.

The classical approach for tracking is the Kalman Filter (KF) (*Fieguth, 2010*) and its many variations, all based on a predictor-corrector framework, meaning that the filter alternates between prediction (asserting the time-dynamics) and correcting (asserting information based on the measurements). For the inverse problem under study, we consider the current location estimation (filtering) in a two dimensional environment. Synthetic object tracking problems, as considered here, are studied in a variety of object tracking papers (*Kim et al., 2019*; *Choi & Christensen, 2013*; *Black, Ellis & Rosin, 2003*; *Lyons & Benjamin, 2009*), where the specific tracking problem in this section is inspired from the approach of *Fraccaro et al. (2017)*, *Vermaak, Lawrence & Pérez (2003)*.

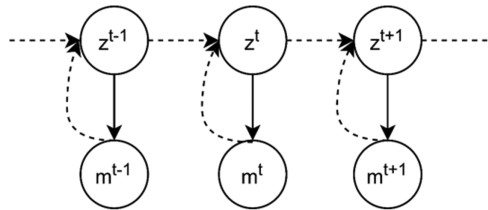

**Figure 13 Graphical model for single object tracking: the goal is to estimate the location of a moving ball in the current frame in a bounded 2D environment. $\underline{m}^t$ denotes the current measured location and $\underline{z}^t$ is the current state.**

The inverse problem task is to estimate the current ball location, given the noisy measurement in the corresponding time step and the previous state of the ball. Formally, we denote the measured ball location by $\underline{m}^t$, and the system state, the current location of the ball, as $\underline{z}^t$. The graphical model in Fig. 13 illustrates the problem definition of the tracking problem, where the objective of the inverse problem is to address the dashed line, the inference of system state from corresponding measurement.

To perform the experiments, we generate the training and test sets similar to *Fraccaro et al. (2017)* except that we assume that our measurements are received from a detection algorithm, which detects the ball location from input images having a size of 32 × 32 pixels, and that the movement of the ball is non-linear.

In each training and test sequence the ball starts from a random location in the 2D environment, with a random speed and direction, and then moving for 30 time steps. The dynamic of the generated data includes changing the ball location $\underline{z}^t$ and its velocity $\underline{v}^t$ as

$$\underline{z}^t = \underline{v}^{(t-1)}\Delta t + \underline{z}^{(t-1)} \tag{37}$$

$$\underline{v}^t = \underline{v}^{(t-1)} - (c(\underline{v}^{(t-1)})^2 \text{sign}(\underline{v}^t)) \tag{38}$$

where $c$ is a constant and is set to 0.001. In our data, collisions with walls are fully elastic and the velocity decreases exponentially over time. In this simulation, the training and testing data-sets contain 10,000 and 3,000 sequences of 30-time steps, respectively.

The training measurement noise is

$$\underline{v} \sim 0.95N(0, 0.2I) + 0.05N(0, 10I), \tag{39}$$

a mixture model of Gaussian noise with 5% outliers. The testing noise is similar,

$$\underline{v} \sim 0.85N(0, 0.4I) + 0.15N(0, 10I) \tag{40}$$

with a higher likelihood of outliers.

The inverse problem is single-target tracking for which the dynamic of the model is unknown. The inverse problem of interest is to find $\underline{z}^t$ in

$$\underline{z}^t = G(\underline{z}^{(t-1)}, m_t) \tag{41}$$

As shown in Fig. 13, we can model our problem as a first order Markov model where the current measurement is independent of others given the current system state. The forward model is then defined as

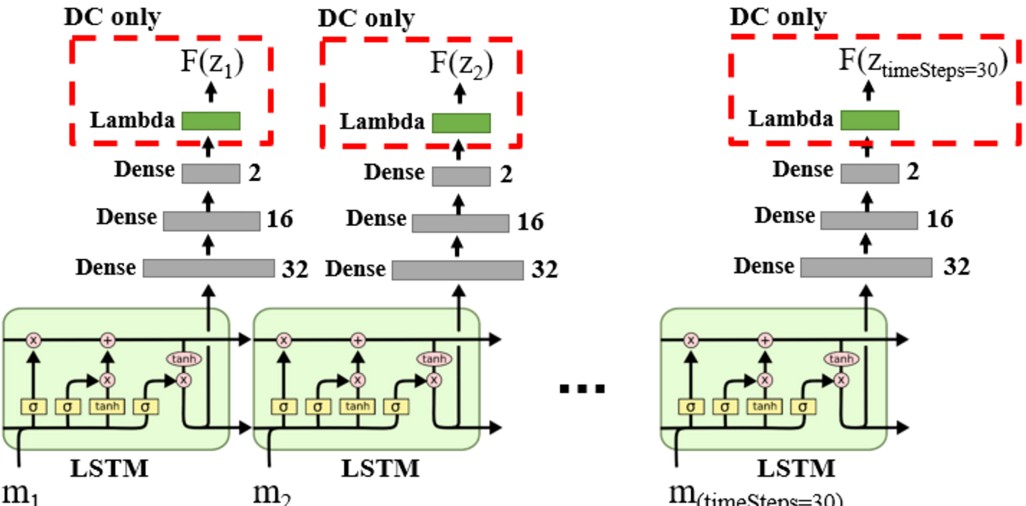

**Figure 14 DNN structure for DM and DC solution categories in the case of single object tracking problem.**

$$\underline{m}^t = F(\underline{z}^t) = C\underline{z}^t + \underline{v}, \quad C = I, \quad \underline{v} \sim N(0, \sigma) \tag{42}$$

We can model Markov models using Recurrent Neural Networks (RNN) (*Krishnan, Shalit & Sontag, 2017*; *Hafner et al., 2019*; *Rangapuram et al., 2018*; *Coskun et al., 2017*). The DNN structure for DM and DC solution categories is shown in Fig. 14, in which the LSTM layers lead the learning process to capture the time state and dynamic information in the data sequences.

We design the regularizer of the DR category as a classifier to classify location feasibility—those locations lying within the border of the 2D environment. Figure 15 shows the positive and negative samples which we used to train the DR regularizer.

As before, we used GA (DR-GA) and Nelder-Mead (DR-NM) algorithms as optimizers for DR. In the case of using Nelder-Mead, the results vary as a function of starting point $\underline{z}_0$, and found that using the last sequence measurement as the starting point empirically gave the best result for DR-NM.

### Visual and numerical results and statistical analysis

Table 9 includes the numerical results obtained by each method in our experiments, where we report the average RMSE between reference and predicted points on the test trajectory as the evaluation criterion for each method.

The obtained results and their statistical analysis are shown in Tables 9 and 10, based on which we conclude that

- In the case of single object tracking, for which system parameters are permitted to evolve and be measured over time (*Fieguth, 2010*), the DM category achieves the best performance using all types of training data. The results are improved when the training data contain representative noise and outliers.

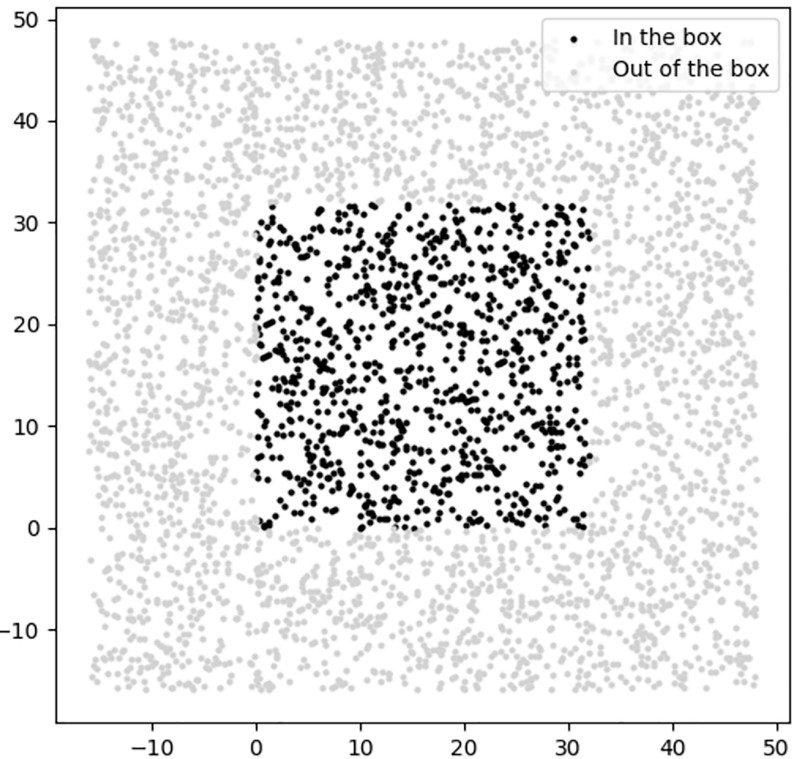

**Figure 15** The positive and negative samples used for training the DR regularizer, where the black and gray samples are in the positive and negative classes, respectively.

**Table 9 RMSE obtained by deep learning solution categories for tracking.** The test data include both noise and outliers.

| Training data | Noise-free | | | | Noisy | | | | Noisy + Outlier | | | |
|---|---|---|---|---|---|---|---|---|---|---|---|---|
| Method | DM | DC | DR-GA | DR-NM | DM | DC | DR-GA | DR-NM | DM | DC | DR-GA | DR-NM |
| RMSE | **1.70 ± 0.05** | 1.79 ± 0.21 | 2.05 ± 0.00 | 1.85 ± 0.00 | **1.72 ± 0.08** | 2.04 ± 0.28 | 2.05 ± 0.00 | 1.85 ± 0.00 | **0.39 ± 0.03** | 1.94 ± 0.02 | 2.05 ± 0.00 | 1.85 ± 0.00 |

Note:
The best result is marked in bold.

**Table 10 Pairwise *p* values for tracking: the Wilcoxon signed rank test checks whether the obtained results are significantly different.**

| *p*-value (Wilcoxon Test) | Training data: Noise-free | | | | Training data: Noisy | | | | Training data: Noisy + Outlier | | | |
|---|---|---|---|---|---|---|---|---|---|---|---|---|
| | DM | DC | DR-GA | DR-NM | DM | DC | DR-GA | DR-NM | DM | DC | DR-GA | DR-NM |
| DM | – | 0.160 | 0.002 | 0.002 | – | 0.002 | 0.002 | 0.002 | – | 0.002 | 0.002 | 0.002 |
| DC | 0.160 | – | 0.013 | 0.130 | 0.002 | – | 0.322 | 0.027 | 0.002 | – | 0.002 | 0.002 |
| DR-GA | 0.002 | 0.013 | – | 0.002 | 0.002 | 0.322 | – | 0.002 | 0.002 | 0.002 | – | 0.002 |
| DR-NM | 0.002 | 0.130 | 0.002 | – | 0.027 | 0.002 | 0.002 | – | 0.002 | 0.002 | 0.002 | – |

- When the training does not include outliers, the DR-NM category achieves the second rank after DM; note that DR-NM is an unsupervised framework without a learning phase, showing that a learning phase is not necessarily required, and that looking only into test cases can give reasonable results.

**Table 11 Performance comparison by solution category and inverse problem types.** Note that $a > b$ means that method $a$ is statistically significantly better than method $b$.

| Inverse problem | Problem type | Training data | Test data | Score (Larger is better) |
|---|---|---|---|---|
| Linear Regression | Reconstruction | Noise-free | Noisy + Outlier | DC > (DR-GA = DR-NM) > DM |
| | | Noisy + Outlier | Noisy + Outlier | (DM = DC) > (DR-GA = DR-NM) |
| 3D Shape Inverse Rendering | Reconstruction | Noise-free | Noisy | DM = DC = DR-GA = DR-NM |
| | | Noise-free | Noisy + Outlier | DC > (DR-GA = DR-NM) > DM |
| | | Noisy | Noisy | DM = DC = DR-GA = DR-NM |
| | | Noisy | Noisy + Outlier | DM > (DC = DR-GA = DR-NM) |
| | | Noisy + Outlier | Noisy | DM > DC > (DR-GA = DR-NM) |
| | | Noisy + Outlier | Noisy+ Outlier | (DM = DC) > (DR-GA = DR-NM) |
| Image Denoising | Restoration | Noisy + Outlier | Noisy+ Outlier | DM > DC > (DR-GA = DR-NM) |
| Single Object Tracking | Dynamic Estimation | Noise-free | Noisy + Outlier | (DM = DC) > DR-NM > DR-GA |
| | | Noisy | Noisy + Outlier | DM > DR-NM > (DC = DR-GA) |
| | | Noisy + Outlier | Noisy + Outlier | DM > DC > DR-NM > DR-GA |
| Burgers' PDE coefficients | Inverse Problems in PDEs | Noise-free | Noisy + Outlier | DC > DM > (DR-GA = DR-NM) |
| | | Noisy | Noisy + Outlier | DC > DM > (DR-GA = DR-NM) |
| | | Noisy + Outlier | Noisy + Outlier | DC > (DR-GA = DR-NM) > DM |

- When the training data include noisy and outlier samples, the solutions' behaviour for single object tracking is similar to that of restoration problems. In particular, in single object tracking the measurements and system parameters are in the same space, like restoration problems.

- In the case of DR solution category for dynamic estimation problems, it is observable that, unlike reconstruction problems, the NM optimization scheme performs better than the GA approach, emphasizing the importance of exploitation power (*Eftimov & Korošec, 2019*; *Xu & Zhang, 2014*), referring to the ability of an optimization method to concentrate on a specific region of the solution space.

## DISCUSSION

Based on the preceding experiments, Table 11 summarizes the overall findings, from which we conclude the following:

- Overall, the presence or absence of outliers in the training phase leads to distinct differences in robustness. Generally, DM will be the best method when the training data does include outliers, whereas DC outperforms other methods if the training does *not* include outliers, based on having a data consistency term in its objective.

- In reconstruction problems, comparing GA and NM optimization approaches in DR shows that GA achieves better performance, indicating the importance of exploration power in optimization for this class of problems.

- The restoration inverse problems, which recover the system parameters from measurements of the same space, need label information (as in DM) to be robust against noise and outliers.

- In the case of restoration problems in static estimation, DM has the highest rank among tested methods. We believe this is because in the process of finding a mapping from one space to itself, the exploitation of accurate solution matters and this property is achieved using label information in the process of training the framework.

- In the case of dynamic estimation problems, the DR solution performs well when the training data do not include outlier samples. Therefore we conclude that this class of problems could be solved without needing a learning phase and that solely the test case is sufficient to find a robust solution.

- The dynamic estimation problems have additional challenges stemming from the time-dependent state information to be captured, an attribute which leads the solution to have different behavior from other problem types. We observed that there are similarities, based on the measurement and system parameter spaces, between the robustness power of the solution categories' performance in a dynamic estimation problem and a static estimation problems with the same measurement and system parameter spaces.

- For PDE inverse problems, the DC solution category achieves the best performance among the methods, and it is the DC learning phase which plays an important role in its performance.

## CONCLUSIONS

This paper investigated deep learning strategies to explicitly solve inverse problems. The literature on deep learning methods for solving inverse problems was classified into three categories, each of which was evaluated on sample inverse problems of different types. Our focus is on the robustness of different categories, particularly with respect to their handling of noise and outliers. The results show that each solution category has a different behavior in the sense of strengths and weaknesses with regards to problem assumptions, such that the problem characteristics need to be considered in selecting an appropriate solution mechanism for a given inverse problem.

Typically, reconstruction problems need more exploration power and the existence of outliers in their training data makes DM the most robust among deep learning solution categories. Otherwise, when the training data do not include outliers for reconstruction problems, DC achieves the best performance, although not using label information in its training phase. The restoration problems need a greater degree of exploitation power for which the DM methods are best suited. In the case of dynamic estimation problems, when the training data do not include outliers, DR achieves second rank, indicating that dynamic estimation problems can be solved with reasonable robustness without a need for learning in the presence of noise. The solution categories for inverse problems in PDEs have specific strategies in the literature, for which the DC category shows the best performance among for almost all types of training data.

### Funding
The authors received no funding for this work.

### Competing Interests
The authors declare that they have no competing interests.

### Author Contributions
- Shima Kamyab conceived and designed the experiments, performed the experiments, analyzed the data, performed the computation work, prepared figures and/or tables, and approved the final draft.
- Zohreh Azimifar conceived and designed the experiments, authored or reviewed drafts of the paper, and approved the final draft.
- Rasool Sabzi conceived and designed the experiments, performed the experiments, performed the computation work, prepared figures and/or tables, and approved the final draft.
- Paul Fieguth conceived and designed the experiments, analyzed the data, authored or reviewed drafts of the paper, and approved the final draft.

### Data Availability
The deep neural networks' codes and their stored weight matrix are available in the Supplemental File.

### Supplemental Information
Supplemental information for this article can be found online at http://dx.doi.org/10.7717/peerj-cs.951#supplemental-information.

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
