# Peer review of "Deep learning methods for inverse problems"

_PeerJ Computer Science, doi:10.7717/peerj-cs.951_

## Round 0.1 · original submission · Major Revisions

The decision was made with the input of the two reviewers. I hope you will take these concerns into consideration, and I look forward seeing the revised paper.

Reviewer 1 ·

Basic reporting

I suggest you clarify the innovation of this paper in the introduction, that is, the contribution made by this article.
Secondly, this article contains a large number of formulas. I hope you can explain the meaning of each symbol in the formula in detail, so as not to cause trouble to readers.
In addition, the figures in this paper need to be explained in more detail or clearer images are selected, as shown in FIG. 11
Formal results should include clear definitions of all terms and theorems, and detailed proofs.

Experimental design

This article explores deep learning methods for inverse problems. In order to facilitate readers to implement the methods used in this article, I suggest that you need to clearly list the hyperparameters required for deep learning training in the article.
The data set used in this study are available publically which is completely limited and imbalance.
In this article, you use various quantitative indicators to measure the advantages and disadvantages of various methods. I hope you can describe in detail the principle of these indicators and their relationship with the quality of the model.
For the overall beauty of the article, I suggest you use a unified style for the drawings of this article.

Validity of the findings

The methods used in this article are the results of early work. I suggest you use the latest achievements to enrich the work of this article, such as the work of the last year.

Additional comments

No comment. See the comments in the above three parts for details

Reviewer 2 ·

Basic reporting

The main innovation of this manuscript is to compare the robustness of the three categories, and report a statistical analysis of their differences, which is not novel enough. The method of this paper is not innovative enough. In fact, most of the work is done by combining other people’s methods. Authors need to highlight their innovative contributions.
Although the article is clear on the whole, I still hope you pay attention to the clarity of the article. There are numerous places in the article that say the same thing over and over again. I hope you can check and delete the redundant parts.
Pay attention to consistency to avoid confusion. Another obvious problem with this paper is the lack of sufficient experimentation to demonstrate the validity and applicability of the three methods.

Experimental design

I recommend you to read this manuscript: " Quadratic Residual Networks: A New Class of Neural Networks for Solving Forward and Inverse Problems in Physics Involving PDEs"Perhaps the author can find inspiration by reading more literature in this field to further optimize this paper.
Please check the normalization and accuracy of the figure in the paper carefully. Many figures are far away from the analysis paragraphs. I hope they can be adjusted.
The literature on deep learning methods for solving inverse problems was classified into three categories, of which was evaluated on sample inverse problems of different types. Why not try more categories? You can make some experiments on more categories.

Validity of the findings

One obvious problem with this paper is the lack of enough experimentation to demonstrate the validity and applicability of the proposed method. The author needs to do more experiments with more angles and show them in this paper.

Additional comments

No comments

---

## Round 0.2 · accepted · Accept

The revised paper meets the requirements of the reviewers and is suitable for publication.

Reviewer 1 ·

Basic reporting

According to the author's response, the author solved related problems, revised the grammatical errors in the revised paper, and clearly expounded the contribution of this paper.

Experimental design

No comment.

Validity of the findings

No comment.

Additional comments

According to the author's response, the author solved related problems, revised the grammatical errors in the revised paper, and clearly expounded the contribution of this paper.

Reviewer 2 ·

Basic reporting

The article meet our standards. The article include sufficient introduction and background to demonstrate how the work fits into the broader field of knowledge. Most of the problems have been solved.

Experimental design

In last version the article is the lack of sufficient experimentation to demonstrate the validity and applicability of the three methods. The author repeated the training procedures 5-10 times and reported the resulting statistics and aggregated the overall results in a separate “Discussion” section to provide a
insights regarding issues facing all of the types of inverse problems in their experiments.

Validity of the findings

The author maintain that the relative merits and the relative applicability of the various methods is not particularly understood, and it is precisely this relative performance which is the purpose and contribution of this paper.

---

## Author Rebuttal · Round 0.2

**Dear PeerJ Reviewers,**

The authors would like to thank the anonymous reviewers for their valuable comments. Below follow our responses to the comments:

**Reviewer 1**

1) I suggest you clarify the innovation of this paper in the introduction, that is, the contribution made by this article.

Thank you; we added an itemized list for describing our contributions to the Introduction (lines 94-97).

2) Secondly, this article contains a large number of formulas. I hope you can explain the meaning of each symbol in the formula in detail, so as not to cause trouble to readers. Formal results should include clear definitions of all terms and theorems, and detailed proofs.

We have added more detailed descriptions to Equations 3-6. The notation of all formulas in the paper should be consistent, and have been re-checked. For all used theorems, we cite the associated references. Please let us know if further proofs need to be added to the paper.

3) In addition, the figures in this paper need to be explained in more detail or clearer images are selected, as shown in FIG. 11

We replaced the images in Fig. 12 (previously 11) with clearer versions. All other figures were checked for clarity.

4) This article explores deep learning methods for inverse problems. In order to facilitate readers to implement the methods used in this article, I suggest that you need to clearly list the hyperparameters required for deep learning training in the article.

Thank you. We added a description about learning rate and number of epochs for training deep in lines 270-274.

5) In this article, you use various quantitative indicators to measure the advantages and disadvantages of various methods. I hope you can describe in detail the principle of these indicators and their relationship with the quality of the model.

Our objective in this paper is to compare the robustness of existing deep learning approaches to solving inverse problems in the presence of noise and outliers. For each problem we formulated a forward model and articulated the strategy which we selected for introducing noise into the training and test data. We have re-checked the paper to ensure the clarity of these approaches.

6) For the overall beauty of the article, I suggest you use a unified style for the drawings of this article.

In the case of DNN diagrams we plot the feedforward layers with black rectangles and convolutional layers with blue cubes, and batch normalization layers with green rectangles. For the linear regression problem we only used feedforward layers, whereas for image denoising we used convolutional DNNs, making the styles seem different. We have re-checked the figures to ensure a uniform style.

7) The methods used in this article are the results of early work. I suggest you use the latest achievements to enrich the work of this article, such as the work of the last year

Thank you; we added a new class of inverse problems, *inverse problems in Partial Differential Equations (PDE)* (Lines 186-189 , new Section 0.5 in experiments,…) for which there are recent solutions proposed in recent years.

**Reviewer 2**

1) The main innovation of this manuscript is to compare the robustness of the three categories, and report a statistical analysis of their differences, which is not novel enough. The method of this paper is not innovative enough. In fact, most of the work is done by combining other people's methods. Authors need to highlight their innovative contributions.

The objective of this paper is to categorize existing deep learning solutions for inverse problems, in terms of their objective criterion.  In this sense the reviewer is correct:  the paper is examining network methods which have been proposed by other authors.  However we would maintain that the relative merits and the relative applicability of the various methods is not particularly understood, and it is precisely this relative performance which is the purpose and contribution of this paper.

2) Although the article is clear on the whole, I still hope you pay attention to the clarity of the article. There are numerous places in the article that say the same thing over and over again. I hope you can check and delete the redundant parts.

We rechecked the article structure and have improved clarity and removed redundant parts.

3) Another obvious problem with this paper is the lack of sufficient experimentation to demonstrate the validity and applicability of the three methods.

We performed statistical analyses of the obtained numerical results to validate the observations. We repeated the training procedures 5-10 times and reported the resulting statistics. We aggregated the overall results in a separate "Discussion" section to provide a insights regarding issues facing all of the types of inverse problems in our experiments.

4) I recommend you to read this manuscript: "Quadratic Residual Networks: A New Class of Neural Networks for Solving Forward and Inverse Problems in Physics Involving PDEs". Perhaps the author can find inspiration by reading more literature in this field to further optimize this paper.

We thank the reviewer for this suggestion. On this basis, we added a new class of inverse problems, *inverse problems in Partial Differential Equations (PDE)* (Lines 186-189 , new Section 0.5 in experiments,…) , which we now compare in our experiments. We also reviewed recent solution categories and have updated the "Literature review" section.

5) Please check the normalization and accuracy of the figure in the paper carefully. Many figures are far away from the analysis paragraphs. I hope they can be adjusted.

Yes, we will move the figures closer to their respective analysis. We have also replaced the images in Figure 12 with clearer ones.

6) The literature on deep learning methods for solving inverse problems was classified into three categories, of which was evaluated on sample inverse problems of different types. Why not try more categories? You can make some experiments on more categories.

Thank you for this suggestion. Indeed, as was already mentioned, we added a new class of inverse problem type, Inverse problems in PDEs, to the formulations and experiments of the paper, and also added more reviewed methods to the literature review.